

# Light dark matter annihilation and scattering in LHC detectors

**Martin Bauer[1], Patrick Foldenauer[1], Peter Reimitz[2] and Tilman Plehn[2]**

**1** Institute for Particle Physics Phenomenology, Durham University, United Kingdom
**2** Institut für Theoretische Physik, Universität Heidelberg, Germany

## Abstract

We systematically study models with light scalar and pseudoscalar dark matter candidates and their potential signals at the LHC. First, we derive cosmological bounds on models with the Standard Model Higgs mediator and with a new weak-scale mediator. Next, we study two processes inspired by the indirect and direct detection process topologies, now happening inside the LHC detectors. We find that LHC can observe very light dark matter over a huge mass range if it is produced in mediator decays and then scatters with the detector material to generate jets in the nuclear recoil.

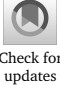

## Content

# 1 Introduction

Discovering the nature of dark matter remains one of the most exciting and pressing goals of fundamental physics. Unfortunately, the mass of the dark matter agent is not known. As a starting point one can distinguish between dark matter with large momenta compared to the local energy density, behaving like particles, and dark matter with small momenta compared to the local energy density which leads to a high occupation number and can be described as a classical wave. We know that dark matter is non-relativistic with a velocity fixed by the escape velocity of galaxies, so dark matter momentum translates directly into masses. Relatively light dark matter can be fermionic or bosonic, while dark matter with masses below few 100 eV can only be bosonic because of the Pauli exclusion principle. It is then referred to as ultralight dark matter (ULDM) or fuzzy dark matter. Typically, particle and wave dark matter require very different search strategies with different reach and limitations.

Searches for particle dark matter with direct detection experiments have limited mass reach due to recoil energy thresholds [1] and for indirect detection backgrounds become very challenging below the soft X-ray spectrum scale as evidenced by the controversy around the 3.5 keV line [2]. Collider searches for dark matter are different from both direct and indirect detection experiments in that they are sensitive to the mediator mass, which determines the missing energy and are largely independent of the dark matter mass as long as the mediator can decay on-shell. If dark matter is lighter than roughly keV and the mediator is not heavy enough to produce a large missing energy signal ($\lesssim 100$ GeV), it could have significant interactions with SM particles and still escape established search strategies for particle dark matter. New experimental techniques aim to extend direct detection probes to smaller masses [3], by utilizing electron recoils [4–10], beta decays and nuclear absorption [11], atomic excitations [12], or absorption in superconductors and other systems with small gaps [6, 13, 14]. Some of these projections promise sensitivity down to the eV scale. Below this mass scale, wave dark matter leads to the variation of fundamental constants, which can be probed in atomic spectroscopy [15–18], laser interference [19], Eot-Wash and ULDM-fifth-force experiments [20, 21]. Other new strategies include nuclear clocks [22], compact binary systems [23, 24] or gravitational wave detectors [25–27]. All these measurements probe masses down to $10^{-24}$ eV, below which the de Broglie wavelength is larger than 100 kpc and galaxy-size structures do not form [28].

There exist a few reasons to favor wave dark matter over particle dark matter, because the macroscopic de Broglie wave length can suppress the formation of small structures and lead to less cuspy halo profiles as opposed to cold dark matter [29–33]. For dark matter that only interacts gravitationally, there is a narrow, preferred mass scale of $10^{-22}$ eV which suppresses kpc-sized cusps and substructures. If there are additional *repulsive* interactions, wave dark matter can solve these problems for a range of masses over many orders of magnitude [34–36]. Even though this wave dark matter can be described classically, it is clear that there has to exist a description in terms of quantum field theory.

In this paper we provide an overview of many constraints on ULDM, formulated in terms of particle physics models. A proper description of these models in terms of a quantum field theory might not be necessary to derive certain implications of wave dark matter, but developing such models without a microscopic theory is overly naive[1]. In Sec. 2 we define a set of models for scalar and pseudoscalar dark matter and derive their respective constraints from cosmology and low energy physics. When we add such a light dark matter field to the Standard Model Lagrangian and study constraints, we need to define a mediator. An obvious choice is the Standard Model Higgs, such that the light dark matter field becomes a minimal extension of the scalar sector of the Standard Model. However, there is always the chance that this mediator is a new scalar particle of an extended Higgs sector. While we do not claim that these

---

[1]In short, quantizing fields relevant for cosmology is a matter of physics honor, not of necessity.

scalar and pseudoscalar models give a comprehensive coverage of the light dark matter model space [37–39], they do provide representative cases for further studies.

In Sec. 3 we then argue that collider searches can provide competitive and complementary sensitivity reach for light and ultra-light dark matter. We consider two processes, inspired by the process topology of *direct and indirect detection,* but now applied to interactions with the LHC detectors. First, ULDM could be produced at large boost and annihilate with the local dark matter density in ATLAS or CMS. This would lead to the production of pairs of photons or electrons, in analogy to indirect detection signals. Unfortunately, we find that ATLAS cannot simply play the role of the galactic center and provide a discovery. Second, ULDM can again be produced for instance in Higgs decays and then inelastically scatter off the detector material. In analogy to direct detection it will leave displaced jets which are extremely unlikely to appear as Standard Model backgrounds. Such *displaced recoil jets* can generally appear in the first dense layers of the LHC detectors, which for ATLAS will be the two calorimeters and for CMS can include the silicon tracker. They are fundamentally different from other displaced signatures such as displaced vertices from decaying massive particle or emerging jets [40]. Unlike classic direct detection this signal does not probe the actual dark matter nature of the candidate particle. Nevertheless, we find that this strategy is promising and propose a new signature of mono-jets and displaced jets in dense detector components. This kind of signature can obviously be extended to other models with light and very weakly interacting particles produced at the LHC, like sterile neutrinos.

## 2   Light dark matter and cosmology

Cold dark matter with a non-relativistic velocity of $v/c \sim 10^{-3}$ and mass $m$ has a local number density of $n = (0.04\,\text{eV})^4/m$ and a de Broglie wavelength of $\lambda_{\text{dB}} = 2\pi/(mv)$. The occupation number then scales like

$$n\,\lambda_{\text{dB}}^3 \approx 6.35 \cdot 10^5 \left(\frac{\text{eV}}{m}\right)^4 , \tag{1}$$

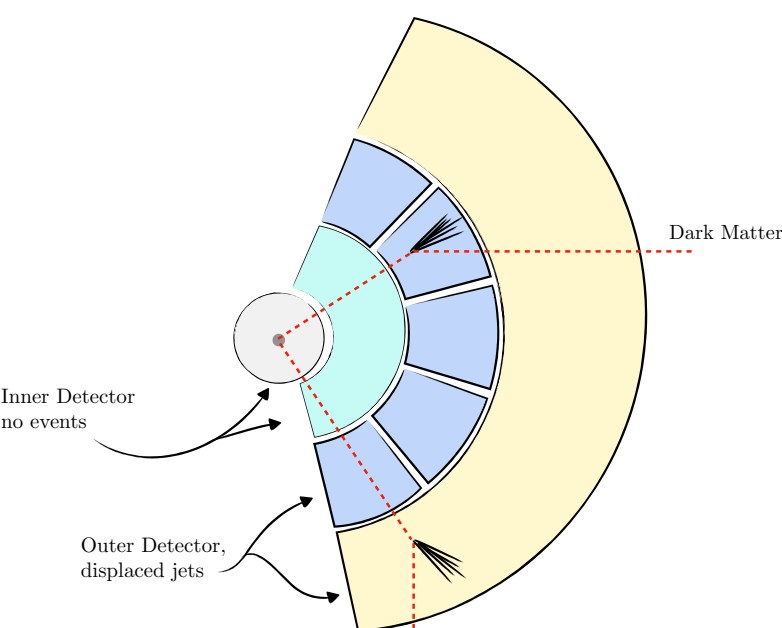

Figure 1: Illustration of the displaced recoil jet signature.

which is huge for $m \lesssim 1$ eV, so dark matter in this regime can be treated as a classical field [41]. In this mass range dark matter can only be bosonic and the correct relic density can be explained by the misalignment mechanism [42, 43] or by the Affleck-Dine mechansim if the scalar sector has an explicitly broken global symmetry [44]. In this regime the only possible decay is into photons and the decay width is suppressed by the dark matter mass. As an example, a dark matter scalar can couple to two powers of the QED field strength $F_{\mu\nu}$ such that its life time scales like $m^3/\Lambda^2$. Similarly, a light vector can couple to three powers of the field strength, leading to a life time proportional to $m^9/\Lambda^8$. Depending on the suppression scale, this implies that light dark matter does not require an additional symmetry to make it stable for $\Lambda \gtrsim 10^4 \, \text{GeV}(m/\text{eV})^{3/2}$ and $\Lambda \gtrsim 2.25 \, \text{keV}(m/\text{eV})^{9/8}$, respectively [45]. This is particularly true for the QCD axion for which the mass is related to the suppression scale, $m \approx 5.7 \cdot 10^{-6} \, \text{eV}(10^{12} \, \text{GeV}/\Lambda)$ [46]. Stability becomes a bad assumption for smaller suppression scales $\Lambda$ or masses above the electron-positron threshold $m > 2m_e$.

For dark matter with negligible self-interactions, the balance between gravitational attraction and quantum pressure determines the scale for which stable structures form and for $m \approx 10^{-22}$ eV the quantum pressure suppresses kpc structures [30]. This opens a solution to the small scale problems of cold particle dark matter, providing a better fit to the matter power spectrum at small scales [47]. In the presence of repulsive self-interactions, masses below the eV-scale kpc-sized structures can be effectively suppressed [48–50]. In contrast, attractive self-interactions have a destabilizing effect and lead to collapsing structures such as boson stars [51, 52]. Axions lead to trigonometric potentials which always induce an attractive self-interaction [34]. Nevertheless, the focus of light dark matter models has been on axion-like candidates inspired by string theory [53–55]. Given the existing constraints they require a very low explicit breaking scale associated with a new physics sector well below the GeV scale even for Planck suppressed couplings [56]. We discuss models with a $Z_2$-symmetry with repulsive self-interactions and point out interesting consequences for low-energy and cosmological constraints in a class of models with derivative couplings and $Z_2$-symmetry. This symmetry guarantees dark matter stability for the whole mass range and self-interactions can be repulsive. In our overview we distinguish two different classes: scalar and Goldstone-boson (axion-like) dark matter coupled through the Higgs portal as well as through a scalar portal with a new mediator.

**Number of effective degrees of freedom**   Before detailing the models under study in this work, we want to give some general remarks on possible constraints on light (pseudo-)scalar DM from the number of effective degrees of freedom $N_{\text{eff}}$ in the early universe.

Depending on the strength of the interaction mediated through the Higgs or scalar portal studied in this paper, dark matter is in thermal equilibrium with the SM thermal bath in the early universe. If this interaction is strong enough to keep the DM in equilibrium until after the QCD phase transition has occurred, it can contribute a significant fraction to the radiation energy density. Such a contribution is typically measured in units of the contribution of a single relativistic neutrino species $\Delta N_{\text{eff}} \equiv \rho(\text{DM})/\rho(\nu)$. The SM predicts three relativistic species in the epoch before recombination $N_{\text{eff}}^{\text{SM}} = 3.046$. This is in agreement with bounds obtained by the Planck collaboration, the strongest one coming from a fit to CMB polarisation, lensing and baryon acoustic oscillation data $N_{\text{eff}}^{\text{SM}} = 2.99^{+0.34}_{-0.33}$ (95% CL) [57], whereas a fit to only the power spectrum yields $N_{\text{eff}}^{\text{SM}} = 3.00^{+0.57}_{-0.53}$. This assumes the standard $\Lambda$CDM model and depends significantly on other underlying parameters, *e.g.* the helium abundance during BBN [58]. In BSM scenarios, *e.g.* if neutrinos are allowed to decay [59, 60] or if a Majoron heats the neutrino sector [61], the value of the fit can change considerably. The contribution of (pseudo-)scalar dark matter is $\Delta N_{\text{eff}} \approx 2$ in the case of late decoupling, $T_{\text{dec}} \lesssim 1$ MeV, whereas earlier decoupling leads to contributions of $\Delta N_{\text{eff}} \lesssim 0.5$ for $T_{\text{QCD}} > T_{\text{dec}} \gtrsim 1$ MeV,

and $\Delta N_{\text{eff}} \lesssim 0.05$ for $T_{\text{dec}} \gtrsim T_{\text{QCD}}$ [62,63]. It is therefore safe to assume that DM decoupling before the QCD phase transition is currently unconstrained by data. For DM decoupling after the QCD phase transition, but at $T_{\text{dec}} > 1$ MeV, the contribution to $\Delta N_{\text{eff}}$ varies between 0.05 and 0.5. Depending on the precise decoupling temperature and the specifics of the dark sector we consider this value unfavoured but not excluded by Planck. A future measurement by the Simons Observatory [64] or CMB-S4 [65] with their respective projected sensitivities of $\sigma(N_{\text{eff}}) = 0.05$ and $\sigma(N_{\text{eff}}) = 0.03$ would firmly exclude these scenarios.

The models we are studying in this work require large couplings to gluons. As a consequence the DM will be kept in equilibrium with the SM until after the QCD phase transition.[2] However, once the temperature drops below $T_{\text{dec}} \lesssim m_\pi$, the dark sector is in contact with the SM thermal bath only via DM DM $\leftrightarrow e^+e^-$ or DM DM $\leftrightarrow \gamma\gamma$ scattering,[3] which are strongly suppressed in all models we are considering. We therefore assume that it is always possible to decouple in the window $m_\pi > T_{\text{dec}} \gtrsim 1$ MeV, where the contribution to $\Delta N_{\text{eff}}$ is still not excluded by Planck.

## 2.1 Scalar dark matter

A scalar singlet $s$ protected by a $Z_2$-symmetry provides a UV-complete model for light dark matter

$$\mathcal{L} \supset \frac{1}{2}\partial_\mu s \partial^\mu s - \frac{1}{2}m_s^2 s^2 - \frac{1}{4!}\lambda_s s^4. \tag{2}$$

Vacuum stability requires $\lambda_s \geq 0$, which implies repulsive self-interactions. Renormalizable couplings to the SM can be established through the Higgs portal

$$\mathcal{L} \supset -\frac{1}{2}\lambda_{hs} s^2 H^\dagger H. \tag{3}$$

For the SM Higgs boson we remind ourselves of the effective coupling to gluons and photons, derived from the low-energy Lagrangian

$$\mathcal{L} \supset \frac{g_{h\gamma\gamma}}{v} h F_{\mu\nu}F^{\mu\nu} + \frac{g_{hgg}}{v} h \operatorname{Tr} G_{\mu\nu}G^{\mu\nu}, \tag{4}$$

with $g_{hgg} = \alpha_s/(12\pi)$ and $g_{h\gamma\gamma} = -47\alpha/(72\pi)$ [66] in the consistent heavy top limit and integrating out the $W$-boson at one loop. Higgs-induced dark matter self-interactions can be large for sizable $\lambda_{hs}$, but any contribution can be absorbed by choosing appropriate values of $\lambda_s$. Scalar dark matter with a Higgs portal is effectively a two parameter model, and both $\lambda_{hs}$ and $m_s$ need to be independently very small to have a viable ultra-light dark matter candidate.

As an alternative, we consider the Lagrangian of Eq.(2) without a Higgs portal, but with a new scalar mediator $\phi$ and an effective coupling to gluons

$$\mathcal{L} \supset -\frac{1}{2}m_\phi^2 \phi^2 - \frac{\mu_{\phi s}}{2}\phi s^2 - \frac{\alpha_s}{\Lambda_\phi}\phi \operatorname{Tr} G_{\mu\nu}G^{\mu\nu}. \tag{5}$$

In contrast to the Higgs portal model, the mediator model introduces three additional parameters, the mediator mass $m_\phi$, the dimensionful coupling strength to dark matter $\mu_{\phi s}$ and a coupling to gluons suppressed by the scale $\Lambda_\phi$. For both of these models we need to consider a set of low-energy and cosmological constraints. We will collect them in Fig. 2 for the Higgs portal and in Fig. 3 for the scalar mediator. In the case of the mediator model we usually fix $m_\phi = 100$ GeV.

---

[2]We want to take this opportunity to thank Simon Knapen for pointing this out to us.

[3]In the case of a scalar mediator and $T_{\text{dec}} < T_{\text{QCD}}$ with couplings only to gluons, dark matter can also annihilate into photons through virtual pion intermediate states DM DM $\to \pi^0\pi^0 \to 4\gamma$.

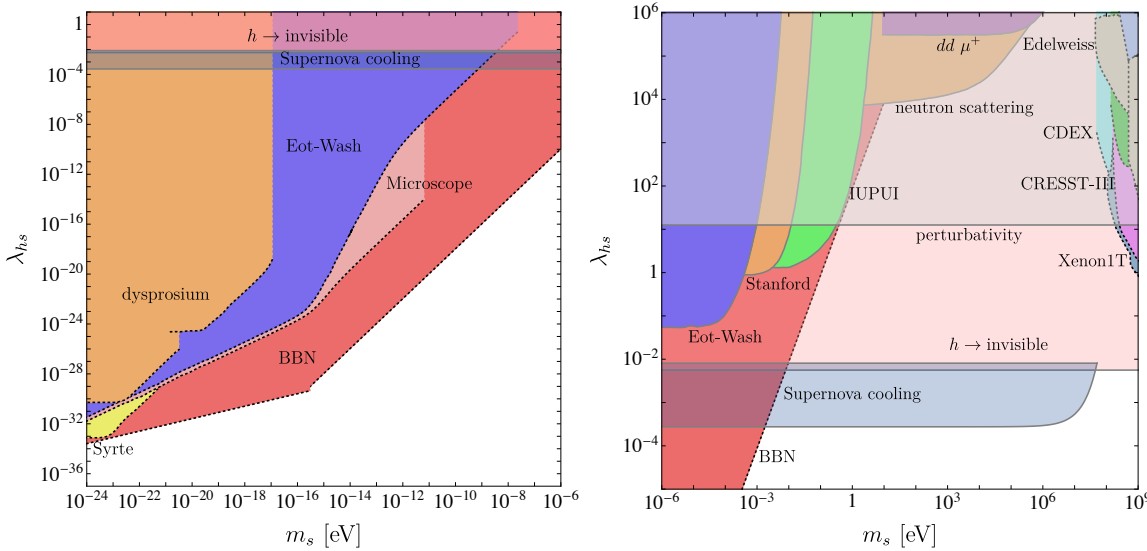

Figure 2: Constraints from precision experiments, cosmology, and direct detection on scalar ULDM with a Higgs portal. The dark matter mass $m_s$ and the portal coupling $\lambda_{hs}$ are the only free parameters. Constraints which require the dark matter nature are shown with dotted contours. The perturbativity bound of $\lambda_{hs} = 4\pi$ is shown in shaded grey.

**Low-energy constraints** The exchange of very light dark matter pairs between nuclei creates a potential that affects precision measurements of low-energy observables. Constraints from neutron scattering, fifth-force searches, Eot-Wash experiments and molecular spectroscopy measurements can be adapted to our fundamental models from Ref [17], where they are reported in terms of effective couplings to nucleons. In our case, the mediator coupling to the gluon coupling induces an effective mediator-nucleon interaction the same way it does for the Higgs [67],

$$\mathcal{L} \supset g_{\phi NN}\, \bar{N}N\, \phi\,, \tag{6}$$

where the effective coupling in terms of the QCD partons is

$$g_{\phi NN} = \frac{8\pi}{11 - \frac{2}{3}n_L}\, \frac{m_N}{\Lambda_\phi}\,, \tag{7}$$

and $n_L$ denotes the number of light quarks. Combined with $\mu_{\phi s}$ this coupling induces a contact interaction of two DM scalars $s$ with two nucleons $N$. We can integrate out the Higgs as well as the scalar mediator using a matching condition of the kind

$$\frac{g_{\phi NN}\, \mu_{\phi s}}{m_\phi^2}\, \bar{N}N\, \frac{s^2}{2} = \frac{1}{\Lambda}\, \bar{N}N\, \frac{s^2}{2}\,, \tag{8}$$

and formulate limits in both scalar dark matter models in terms of

$$\mathcal{L} \supset c_{sNN} s^2 \bar{N}N\,, \tag{9}$$

with the dimensionful coefficients

$$
\begin{aligned}
c_{sNN} &= \lambda_{hs}\, \frac{m_N}{m_h^2}\, \frac{2n_H}{3(11 - \frac{2}{3}n_L)} && \text{(Higgs portal)} \\
c_{sNN} &= \frac{\mu_{\phi s}}{\Lambda_\phi}\, \frac{m_N}{m_\phi^2}\, \frac{8\pi}{11 - \frac{2}{3}n_L} && \text{(scalar mediator)}\,.
\end{aligned}
\tag{10}
$$

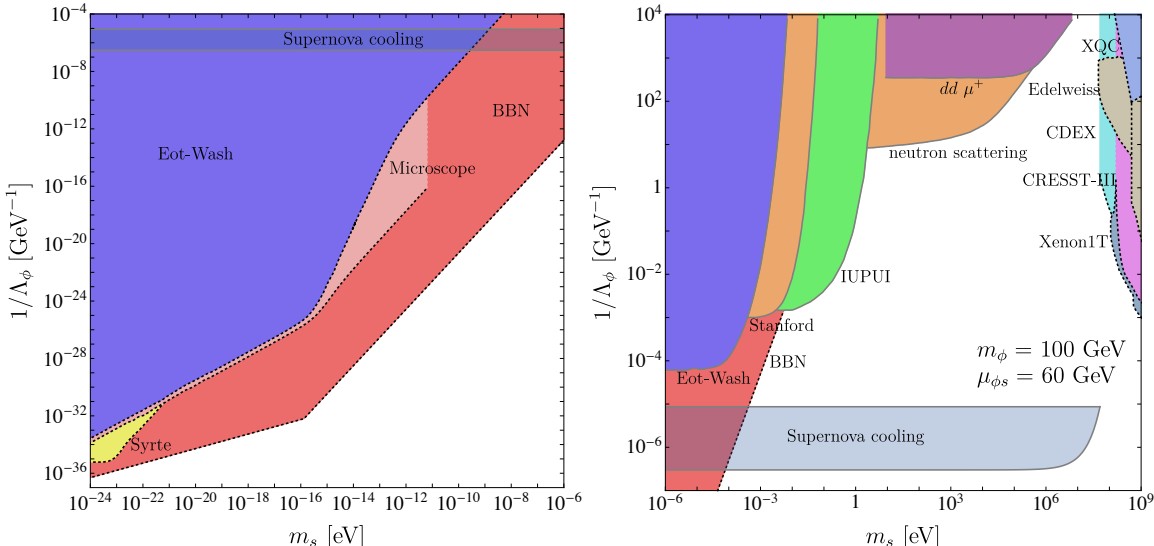

Figure 3: Constraints from precision experiments, cosmology, and direct detection on scalar ULDM with a scalar mediator. In addition to the dark matter mass $m_s$ and mediator-gluon coupling $1/\Lambda_\phi$ we fix the mediator mass to $m_\phi = 100$ GeV its coupling to the dark matter agent to $\mu_{\phi s} = 60$ GeV. Constraints which require the dark matter nature are shown with dotted contours.

These operators lead to a new, fifth force, mediated by the light scalar $s$ that is independent of its dark matter character. The low-energy limits are shown in Fig. 2 as a function of the dark matter mass and the Higgs portal coupling. Eot-Wash limits on fifth forces based on data from torsion balance experiments [68–70] are shown in blue. For relatively high masses, shown in the right panel of Fig. 2, they lose sensitivity for masses $m_s \gtrsim 10^{-4}$ eV, corresponding to around $10^{-4}$ m, the length scale tested by the experiment. Searches for fifth forces with planar geometry are less sensitive, but can probe distances down to a few $\mu$m. The limits from experiments by the Stanford group [71] and at Indiana-Purdue (IUPUI) [72] are shown in the right panel of Fig. 2 in orange and green, respectively. In constrast, the MICROSCOPE satellite [73] tests this interaction in the presence of DM-background and probes deviations in the orbits of test masses. These limits, shown in light red in Fig. 2, are strong up to $m_s \approx 10^{-12}$ eV or length scales around $2 \cdot 10^5$ m, roughly corresponding to the orbit of the satellite [20]. The same DM-background also leads to improved Eot-Wash limits in the same mass range. Below the $\mu$m scale, constraints can be set by neutron-nucleon scattering [74]. The corresponding limit is shown in orange in the right panel of Fig. 2. Finally, molecular spectroscopy experiments are sensitive to forces below the keV scale or $10^{-10}$ m [17]. The strongest limit is provided by measurements with muonic molecular deuterium ions [75] shown in purple in the right panel of Fig. 2.

The dark matter halo acts like a classical background field inducing oscillating variations in fundamental constants that would not appear for a fifth force mediator. From a model perspective, searches for a fifth force and variations of fundamental constants from the dark matter halo constrain the same parameter space. Spectroscopy searches are sensitive to time-dependent oscillations of nucleus and electron masses and the fine-structure constant discussed in App. A.1. Since the frequency of these oscillations are related to the mass, $\omega = m_s c^2/\hbar$, the sensitivity of these searches peaks for a mass related to the total measurement time, and the experiment looses sensitivity below the lowest frequency for which one full oscillation can be measured and for frequencies higher than the shortest time between

measurements [76]. The strongest constraint comes from measurements with rubidium and cesium at LNE-SYRTE [20,77], shown in yellow in the left panel of Fig. 2.

**BBN constraints**   Many of the constraints from low-energy experiments discussed above are not specific for dark matter. For light particles forming dark matter the misalignment mechanism leads to time-dependent oscillations, which induces variations in fundamental constants such as the fine-structure constant and masses of fermions and vector bosons, which are specific to dark matter. Big-bang nucleosynthesis (BBN) predicts the yield of $^4$He as $Y_{\text{th}} = 0.24709 \pm 0.00025$ [78], which agrees well with the measured value $Y_{\text{exp}} = 0.245 \pm 0.003$ [79] and any variation is constrained to the range

$$\frac{\Delta Y}{Y} = -0.008458 \pm 0.012183 \,. \tag{11}$$

The $^4$He abundance at the time of BBN can be written in terms of the proton-neutron ratio

$$Y_{\text{BBN}} = \frac{2}{1 + \frac{p_{\text{BBN}}}{n_{\text{BBN}}}} \,, \tag{12}$$

where

$$\frac{n_{\text{BBN}}}{p_{\text{BBN}}} = \frac{n_W e^{-\Gamma_n t_{\text{BBN}}}}{p_W + n_W(1 - e^{-\Gamma_n t_{\text{BBN}}})} \qquad \text{and} \qquad \frac{n_W}{p_W} = e^{-Q_{np}/T_F} \,. \tag{13}$$

The neutron-proton mass difference can be approximated as $Q_{np} = m_n - m_p = m_d - m_u + \alpha \Lambda_{\text{QCD}}$, and the weak freeze-out temperature is given by [80]

$$T_F = \frac{b\, m_W^{4/3} \sin^{4/3}(\theta_W)}{\alpha^{2/3} M_{\text{Planck}}^{1/3}} \approx 0.75 \,\text{MeV} \,. \tag{14}$$

For small values of $\Gamma_n t_{BBN}$ a possible deviation of the Helium yield can be traced as

$$\frac{\Delta Y}{Y} \approx \frac{\Delta (n/p)_W}{(n/p)_W} - \Delta \Gamma_n t_{BBN} \,. \tag{15}$$

It is sensitive to variations of proton and neutron masses, variations of $M_W$ and $M_Z$ as well as of $\alpha$. In the case of the Higgs portal, these variations are universal apart from the loop-induced photon coupling and following the effective operator analysis in Ref. [15], we derive the constraint for the Higgs portal

$$\frac{1}{m_s^2} \frac{\lambda_{\phi s}}{m_h^2} \left[ -0.25(2g_{h\gamma\gamma}) - 3.79 \right] \simeq (-2.6 \pm 3.7) \cdot 10^{-20} \,\text{eV}^{-4} \,, \tag{16}$$

for $m_s \gg 10^{-16}$ eV and

$$\frac{1}{m_s^2} \left( \frac{m_s}{3 \cdot 10^{-16} \,\text{eV}} \right)^{3/2} \frac{\lambda_{hs}}{m_h^2} \left[ -0.25(2g_{h\gamma\gamma}) - 3.79 \right] \simeq (-1.3 \pm 1.8) \cdot 10^{-20} \,\text{eV}^{-4} \,, \tag{17}$$

for $m_s \ll 10^{-16}$ eV.

For the scalar mediator the operators in Eq.(5) lead to a universal correction to the nucleon masses induced by the coupling to gluons, whereas corrections to other fundamental constants are strongly suppressed. The constraint can therefore be expressed as

$$\frac{1}{m_s^2} \frac{4\pi}{9} \frac{\mu_{\phi s}}{\Lambda_\phi} \frac{1}{m_\phi^2} \simeq (4.2 \pm 6.2) \cdot 10^{-21} \,\text{eV}^{-4} \,, \tag{18}$$

for $m_s \gg 10^{-16}$ eV and

$$\frac{1}{m_s^2}\left(\frac{m_s}{3 \cdot 10^{-16}\,\text{eV}}\right)^{3/2}\frac{4\pi}{9}\frac{\mu_{\phi s}}{\Lambda_\phi}\frac{1}{m_\phi^2} \simeq (2.1 \pm 3.1) \cdot 10^{-21}\,\text{eV}^{-4}\,, \tag{19}$$

for $m_s \ll 10^{-16}$ eV. In Fig. 2 we see that for $m_s \lesssim 10^{-3}$ eV the observed ${}^4$He abundance set during BBN is indeed the most stringent constraint. The results for a scalar mediator displayed in Fig. 3 show a similar situation for a fixed mediator mass $m_\phi = 100$ GeV and coupling $\mu_{\phi s} = 60$ GeV.

**Supernova constraints** For masses below the supernova core temperature of $2m_s < T_{\text{SN}} \approx 30$ MeV, scalar dark matter pairs can be radiated off nuclei inside the supernova core. They leave the star and effectively provide a new source of cooling in addition to neutrinos [81]. This is the so called free streaming energy loss bound assuming that the additional particles can escape the supernova freely. The observation of SN1987A puts a limit on the total energy-loss rate per unit mass, volume and time of

$$\Gamma = \varepsilon_x \rho_{\text{core}} < 10^{-14}\,\text{MeV}^5\,. \tag{20}$$

The dominant process for scalar dark matter production in the core is nucleon bremsstrahlung $NN \to NN + ss$ [82]. For the Higgs portal and the scalar mediator this translates into

$$\begin{aligned}\lambda_{hs} &< 2.75 \cdot 10^{-4} &&\text{(Higgs portal)}\,,\\ \frac{\mu_{\phi s}}{\Lambda_\phi} &< 1.8 \cdot 10^{-5}\left(\frac{m_\phi}{100\,\text{GeV}}\right)^2 &&\text{(scalar mediator)}\,.\end{aligned} \tag{21}$$

For very large couplings, the mean free path [83]

$$\lambda = \frac{1}{n_N(r)\,\sigma_{sN \to sN}}\,, \quad n_N(r) = \begin{cases}\dfrac{\rho_{\text{core}}}{m_p} & \text{for } r \leq R_{\text{SN}}\,,\\[2ex] \dfrac{\rho_{\text{core}}}{m_p}\left(\dfrac{R_{\text{SN}}}{r}\right)^m & \text{for } r > R_{\text{SN}}\,,\end{cases} \tag{22}$$

with $m = 3 \dots 7$ and

$$\sigma_{sN \to sN} \simeq \frac{1}{4\pi}c_{sNN}^2\,, \qquad \text{assuming} \quad m_s, E_s \ll m_N\,, \tag{23}$$

becomes smaller than the supernova radius $\lambda \leq R_{\text{SN}}$. In that case the cooling is not efficient because of repeated scattering of dark matter in the star [81,84], so the light scalars start to thermalize. Once the assumption of free-streaming breaks down and the dark matter is trapped in the supernova, it builds up a scalarsphere of radius $r_0 > R_{\text{SN}}$ characterized by the optical depth criterion

$$\tau_s = \int_{r_0}^\infty \lambda^{-1}\mathrm{d}r = \int_{r_0}^\infty \sigma_{sN \to sN}n_N(r)\mathrm{d}r \leq \frac{2}{3}\,, \tag{24}$$

similar to the axiosphere [81,84]. Combined with applying the energy-loss bound of SN1987A on the black-body radiation of the scalarsphere with a temperature profile of [83]

$$T(r) = T_{\text{SN}}\left(\frac{R_{\text{SN}}}{r}\right)^{m/3}\,, \tag{25}$$

we obtain the constraints

$$\lambda_{hs} < 8.1 \cdot 10^{-3} \qquad \text{(Higgs portal)},$$

$$\frac{\mu_{\phi s}}{\Lambda_\phi} < 5.2 \cdot 10^{-4} \left(\frac{m_\phi}{100\,\text{GeV}}\right)^2 \quad \text{(scalar mediator)}. \qquad (26)$$

The effect of this constraints on our model parameters is shown in Figs. 2 and 3. We note that for DM masses close to the core temperature, $m_s \lesssim T_{\text{SN}}$, we have to include a Boltzmann factor $\exp(-2\,m_s/T_{\text{SN}})$ into the energy loss rate to model the temperature dependence [1]. For masses above $m_s > 10^{-3}$ eV the BBN constraints vanish and supernova cooling yields the leading constraints, independent of the couplings and for dark matter masses up to $m_s \sim 50$ MeV.

**Invisible Higgs decays** A largely model-independent bound comes from searches for invisible Higgs decays. The strongest current limit on the branching ratio of Higgs bosons into invisible final states is $\text{BR}(h \to \text{inv}) \lesssim 13\%$ [85]. Minimizing the underlying assumptions and interpreting the LHC limits in an effective theory framework hardly changes this limit [86]. A future high-luminosity run of the LHC (HL-LHC) could improve this limit by an order of magnitude $\text{BR}(h \to \text{inv}) \lesssim 2\%$ [87]. For the Higgs portal the partial decay rate of the Higgs to two scalars is given by

$$\Gamma(h \to ss) = \frac{\lambda_{hs}^2}{8\pi} \frac{v^2}{m_h} \sqrt{1 - \frac{4m_s^2}{m_h^2}}. \qquad (27)$$

Together with the current bound on invisible Higgs decays, this provides an essentially $m_s$-independent limit on the portal coupling of light DM (i.e. $m_s \ll m_h/2$) of

$$\lambda_{hs} < 5.6 \cdot 10^{-3} \quad \text{(current)},$$

$$\lambda_{hs} < 2.1 \cdot 10^{-3} \quad \text{(HL-LHC)}. \qquad (28)$$

This constraint is absent in the case of the scalar mediator portal. In Fig. 2 we see this limit right above the supernova limit.

**Direct detection** Finally, direct detection experiments based on heavy noble gases have a recoil threshold of $\sim 1$ keV, which translates into a sensitivity to dark matter masses of $m \gtrsim 1...10$ GeV. Cryogenic calorimeter experiments can lower the nuclear threshold to $\sim 100$ eV, providing sensitivity down to dark matter masses of $m \gtrsim 100$ MeV. A similar threshold has been obtained by the space based X-ray Quantum Calorimetry Experiment (XQC) which is sensitive to strongly interacting dark matter in this mass range [88]. We use the results from [18, 89] to show the constraints on the parameters of the Higgs portal in Fig. 2. For DM masses of $m_s \gtrsim 100$ MeV we see the leading constraints from Xenon1T [90] (dark grey), CRESST-III [91] (pink), CDEX [92] (cyan), Edelweiss [93] (pale brown) and XQC [88] (pale blue). In Fig. 3 we show the corresponding limits for a scalar mediator.

It is worth pointing out that further bounds exist from cosmic ray propagation [94,95] and cosmology [96–99], which constrain very large DM-nucleus cross sections.

## 2.2 Pseudoscalar dark matter

Unlike scalar dark matter, pseudoscalar or axion-like (ALP) dark matter [100] is described by a non-renormalizable Lagrangian

$$\mathcal{L} \supset \frac{1}{2}\partial_\mu a \partial^\mu a - \frac{m_a^2}{2}a^2 + \frac{\partial_\mu a}{f}\sum_i \frac{c_i}{2}\,\bar{\psi}_i \gamma_\mu \gamma_5 \psi_i$$
$$+ c_G \frac{g_s^2}{16\pi^2}\frac{a}{f}\,\mathrm{Tr}[G_{\mu\nu}\widetilde{G}^{\mu\nu}] + c_W \frac{g^2}{16\pi^2}\frac{a}{f}\,\mathrm{Tr}[W_{\mu\nu}\widetilde{W}^{\mu\nu}] + c_B \frac{g'^2}{16\pi^2}\frac{a}{f}\,B_{\mu\nu}\widetilde{B}^{\mu\nu}\,. \tag{29}$$

All pseudoscalar couplings are suppressed by at least one power of the mass scale $f$. To understand the role of $f$ and compute the couplings to the Higgs sector we consider the UV-complete theory with a complex scalar breaking a global symmetry

$$S = \frac{s+f}{\sqrt{2}}\,e^{ia/f}\,. \tag{30}$$

In this section the scalar mode $s$ is heavy. Its mass is set by $f$, while the mass of the pseudoscalar $a$ is proportional to some explicit breaking of the shift symmetry parameterized by $\mu$, such that $m_a = \mu^2/f$. A conserved $Z_2$-symmetry $S \to -S$ forbids all dimension-5 operators. Dimension-6 operators are introduced by the renormalizable Higgs portal and the kinetic term of the full theory

$$\mathcal{L} \supset \partial_\mu S \partial^\mu S^\dagger + \mu_s^2 S^\dagger S - \lambda_s (S^\dagger S)^2 - \frac{1}{2}\lambda_{hs}\,S^\dagger S\,H^\dagger H\,. \tag{31}$$

They give a scalar mass $m_s = \sqrt{2\lambda_s}f$ and lead to an effective, derivative Higgs portal suppressed by $1/f^2$ [101],

$$\mathcal{L} \supset -\frac{\lambda_{hs}}{2m_s^2}\,\partial_\mu a \partial^\mu a\,H^\dagger H\,. \tag{32}$$

The derivative Higgs-portal can also be induced by a coupling between the complex scalar and the SM through the effective operator

$$\mathcal{L} \supset \frac{(\partial_\mu S)(\partial^\mu S)^\dagger}{\Lambda_{ha}^2}H^\dagger H = \frac{\partial_\mu a\,\partial^\mu a}{2\Lambda_{ha}^2}H^\dagger H = \frac{\partial_\mu a\,\partial^\mu a}{4\Lambda_{ha}^2}(v^2 + 2v\,h + h^2)\,, \tag{33}$$

where we introduce a specific suppression $1/\Lambda_{ha}$ and, in the last step, insert the Higgs field. In principle, there can be a hierarchy of scales $f \gg \Lambda_{ha}$ and we parametrize effects through the derivative Higgs portal by $\Lambda_{ha}$ from now on. We also note that this operator will be generated from the Higgs portal. Alternatively, we can write it as

$$\frac{\partial_\mu a\,\partial^\mu a}{2\Lambda_{ha}^2}H^\dagger H = -\frac{m_a^2 a^2}{4\Lambda_{ha}^2}(v+h)^2 - \frac{a\partial_\mu a}{2\Lambda_{ha}^2}(v\,\partial^\mu h + h\partial^\mu h)\,, \tag{34}$$

where the second term gives rise to a momentum-dependent scalar coupling to ALP pairs.

As a simple generalization of the Higgs mediator model we again consider a model with a new scalar mediator $\phi$. It is defined by the operators

$$\mathcal{L} \supset -\frac{1}{2}m_\phi^2 \phi^2 - \frac{\partial_\mu a\,\partial^\mu a}{2\Lambda_{\phi a}^2}\phi - \frac{\alpha_s}{\Lambda_\phi}\,\phi\,\mathrm{Tr}\,G_{\mu\nu}G^{\mu\nu}\,. \tag{35}$$

The coupling to gluons is described by the same parameter $\Lambda_\phi$ as in the scalar case of Eq.(5). However, unlike in the scalar case this new scale is supplemented with the new physics scale

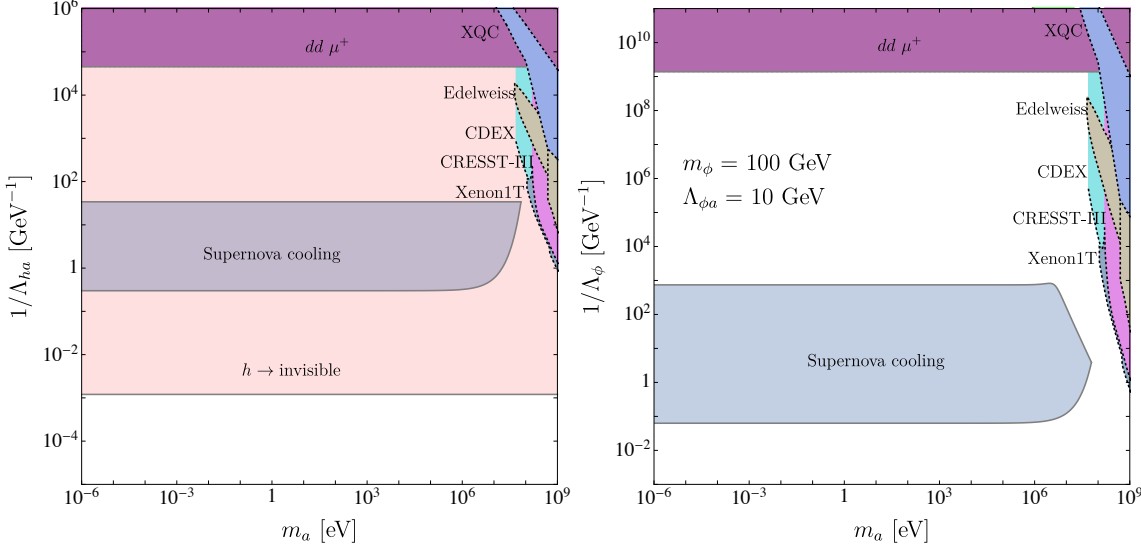

Figure 4: Constraints from precision experiments, cosmology, and direct detector on pseudoscalar or axion-like ULDM with a Higgs mediator (left) and a scalar mediator (right). For the latter we again fix the mediator mass to $m_\phi = 100$ GeV its coupling to the dark matter agent to $\Lambda_{\phi a} = 10$ GeV. Constraints which require the dark matter nature are shown with dotted contours.

of the pseudoscalar $\Lambda_{\phi a}$. Such a scalar portal operator in a $Z_2$-protected symmetry is not very exotic. It has for example been considered to generate a fractional contribution to the effective number of degrees of freedom [101]. More recently, the derivative Higgs portal has been considered in the context of missing energy signals at the LHC in [102].

The operators given in Eq.(33) and (35) induce couplings between ALPs and nuclei,

$$\mathcal{L} \supset c_{aNN} \ \partial_\mu a \ \partial^\mu a \ \bar{N}N \, , \tag{36}$$

with the dimensionful coefficients

$$c_{aNN} = \frac{1}{\Lambda_{ha}^2} \frac{m_N}{m_h^2} \frac{2n_H}{3(11 - \frac{2}{3}n_L)} \quad \text{(Higgs mediator)}$$

$$c_{aNN} = \frac{m_N}{\Lambda_{\phi a}\Lambda_\phi m_\phi^2} \frac{8\pi}{11 - \frac{2}{3}n_L} \quad \text{(scalar mediator)} \, . \tag{37}$$

Constraints from low-energy precision experiments can therefore be discussed in analogy with the bounds on the couplings in Eq.(10). In Fig. 4 we show the constraints adapted from Ref. [17] both for the Higgs mediator and a new scalar mediator. In contrast to the case of the operators in Eq.(29), all ALP interactions mediated by Eq.(32) are momentum suppressed. This generalizes to theories with more than one derivative. The sensitivity of low-energy observables is therefore strongly suppressed with respect to the case of an ALP with linear interactions *and* in contrast to the scalar without a shift symmetry discussed in Sec. 2.1. For a more detailed discussion of this we refer to Appendix A.2. The potential for the long-range force induced by the exchange of at least two ALPs with shift-symmetry, Eq.(36), is suppressed by $1/r^7$ [17], which suppresses the sensitivity from experiments sensitive to effects at large scales. The bounds from Eot-wash experiments and MICROSCOPE are not relevant for the parameter space shown in Fig. 4 and constraints from neutron scattering and molecular spectroscopy provide the dominant low-energy constraints.

**BBN constraints** Constraints on pseudoscalar dark matter from big bang nucleosynthesis are further suppressed by the derivative coupling. Treating DM as a classical field with $a(t) = a_0 \cos(m_a t)$ gives an additional factor of $m_a^2$ that is canceling the mass dependence in Eq.(18). The constraints now read

$$\frac{1}{2\Lambda_{ha}^2 m_h^2}\left(\frac{0.25c_{h\gamma\gamma}}{4\pi} + 3.79\right) \simeq (2.6 \pm 3.7) \cdot 10^{-20} \text{ eV}^{-4} \quad \text{(Higgs mediator)}$$

$$\frac{4\pi}{9\Lambda_\phi \Lambda_{\phi a} m_\phi^2} \simeq (4.2 \pm 6.2) \cdot 10^{-19} \text{ eV}^{-4} \quad \text{(scalar mediator)} \qquad (38)$$

for $m_s \gg 10^{-16}$ eV. We see in Fig. 4 that the derivative interaction weakens the BBN bounds to the point that they do not even appear in the plot.

**Supernova constraints** Similarly, constraints from supernova cooling are strongly suppressed, because the derivatives induce additional temperature suppression in the nuclear bremsstrahlung rate. Using the results from Appendix A.4, we find that couplings in the range

$$\frac{34.1}{\text{GeV}} > \frac{1}{\Lambda_{ha}} > \frac{0.3}{\text{GeV}} \qquad \text{(Higgs mediator)}\,,$$

$$\frac{740}{\text{GeV}} > \frac{1}{\Lambda_\phi}\frac{10\,\text{GeV}}{\Lambda_{\phi a}}\left(\frac{100\,\text{GeV}}{m_\phi}\right)^2 > \frac{0.063}{\text{GeV}} \qquad \text{(scalar mediator)}\,, \qquad (39)$$

are excluded by supernova cooling constraints.

**Invisible Higgs decays** These relatively model-independent constraints work the same way as for the scalar case. The new Higgs decay comes with the partial width

$$\Gamma(h \to aa) = \frac{v^2 m_h^3}{128\pi\Lambda_{ha}^4}\left(1 - \frac{2m_a^2}{m_h^2}\right)^2 \sqrt{1 - \frac{4m_a^2}{m_h^2}} \approx \frac{v^2 m_h^3}{128\pi\Lambda_{ha}^4}\,. \qquad (40)$$

The limits on invisible Higgs decays translate into

$$\Lambda_{ha} \gtrsim 832 \text{ GeV}\,, \qquad (41)$$

for the current bound and $\Lambda_{ha} \gtrsim 1.37$ TeV for the projected bound from the HL-LHC. This reach is clearly limited and an observation would not yield any information on the DM character of a new light particle. Nevertheless, in the left panel of Fig. 4 we see that the invisible Higgs decays give the leading constraint on the model.

**Direct detection** Finally, we again contrast the pseudoscalar model predictions with the limits set by different direct detection experiments. For the Higgs and scalar mediators we see in Fig. 4 that the different experiments systematically probe their respective model parameter space for dark matter masses exceeding $m_a \sim 50$ MeV. Because of the momentum dependence of (36), scattering from the nuclei is suppressed by the dark matter velocity and bounds from direct detection are considerably weaker compared to scalar dark matter. The limits from CRESST [103], XQC [88], Xenon1T [90], CRESST-III [91], CDEX [92] Edelweiss [93] and XQC [88] are shown with the same color coding as in Fig. 3. The constraints seem more important in comparison to the supernova bounds, because the temperature suppression in the latter is more effective than the velocity suppression in ALP-nucleus scattering.

# 3 LHC signatures of light dark matter

To look for a light new particle with a $Z_2$-symmetry and a coupling to the Higgs sector at the LHC, we usually rely on invisible Higgs decays. From Sec. 2 we know that this process has an impressive discovery potential with very few underlying assumptions. However, an observation of invisible Higgs decays would not link the new particle to dark matter. To show such a link we could for instance search for invisible Higgs decays where one of the two light scalar interacts with the DM background and produces two Standard Model states. This process is the LHC equivalent to indirect detection and will be covered in Sec. 3.1. Alternatively, a light scalar produced in Higgs decays can scatter with the detector, a DIS-like process which corresponds to direct detection. We will look at it in Sec 3.2.

Before we study the potential LHC signatures we remind ourselves of the different models defined in Sec. 2:

– scalar $s$ with a Higgs portal, Eq.(3), described by the renormalizable coupling $\lambda_{hs}$;

– scalar $s$ with a mediator $\phi$, Eq.(5). The mediator couples to the dark matter scalar through $\mu_{\phi s}$ and to gluons at dimension five, defining $1/\Lambda_\phi$;

– pseudoscalar $a$ with a dimension-6 coupling to the Higgs given by $1/\Lambda_{ha}^2$, Eq.(33);

– pseudoscalar $a$ with a mediator $\phi$, Eq.(35). The mediator coupling to the pseudoscalar is $1/\Lambda_{\phi a}^2$ and its coupling to gluons defines $1/\Lambda_\phi$;

As discussed before, these four models are subject to a wealth of cosmological constraints.

## 3.1 Dark matter annihilation

If the observed relic density is given by light scalars with a $Z_2$-symmetry, a particle which we produce at the LHC can annihilate with the dark matter background

$$\langle s \rangle s \to \gamma \gamma \,, \tag{42}$$

where $\langle s \rangle$ denotes the dark matter background, also shown in Fig. 5. We assume that pairs of scalars $s$ are produced in Higgs decays and then traverse the dense DM background at high momentum. As an example we stick to the Higgs portal model defined by Eq.(3) throughout this section. In analogy to fixed target experiments the number of photon pairs produced from

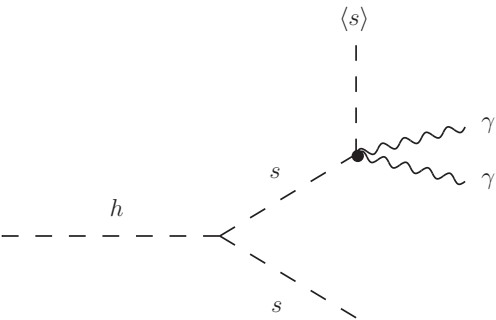

Figure 5: Appearing pair of boosted photons from DM-background scattering. The produced DM scalars originate from a Higgs decay and share the Higgs rest mass between them.

a beam of $N_s$ scalars $s$ with initial energy $E_s$ at a distance $l$ from the production point can be estimated as [104–106]

$$\frac{d^2 N_{\gamma\gamma}}{dE_s\, dl} = N_s\, I_s(E_0, E_s, l)\, \frac{dP_{\text{conv}}}{dl}\,. \tag{43}$$

Here $I_s$ denotes the energy distribution function of the scalars $s$. The number of produced scalars can be calculated as $N_s = N_h\, \text{BR}(h \to ss)$. For simplicity we assume $I_s(E_s, E_0, l) = \delta(E_s - E_0)$, which is justified as long as we do not have significant additional interactions of the scalars with the DM medium. One example for such an interaction would be scalar self-interactions, but they are expected to be small for a DM candidate because of structure formation.

Finally, we are interested in the number of DM-to-photon conversions taking place in the fiducial detector volume. As the scalars move through the DM gas the conversion is described by the usual probability of a single scalar to annihilate with a DM background particle into two photons in a spatial slice $dl$. The differential conversion probability for a single incoming scalar $s$ is given by

$$\frac{dP_{\text{conv}}(l)}{dl} = \frac{e^{-l/\lambda}}{\lambda}\,, \tag{44}$$

with the mean free path

$$\lambda = \frac{1}{n_{\text{DM}}\, \sigma_{\langle s\rangle s \to \gamma\gamma}}\,, \tag{45}$$

in terms of the DM number density $n_{\text{DM}}$. Integrating Eq.(43) along a detector with size $L_{\text{det}}$ at a distance $L_0$ away from the interaction point gives us

$$\begin{aligned}
N_{\gamma\gamma} &= N_s \int_0^{m_h/2} dE_s\; \delta(E_s - m_h/2) \int_{L_0}^{L_0 + L_{\text{det}}} dl'\, \frac{e^{-l/\lambda}}{\lambda} \\
&= N_h\, \text{BR}(h \to ss)\; e^{-L_0/\lambda} \left(1 - e^{-L_{\text{det}}/\lambda}\right),
\end{aligned} \tag{46}$$

where $\lambda$ is evaluated at the Higgs mass scale and the effective Higgs couplings are defined in Eq.(4). The corresponding differential cross section is

$$\frac{d\sigma_{\langle s\rangle s \to \gamma\gamma}}{dt} = \frac{1}{2\pi} \frac{\lambda_{hs}^2\, g_{h\gamma\gamma}^2}{(s - m_h^2)^2}\,. \tag{47}$$

As for all fixed target experiments, the center-of-mass energy is much lower than the momentum of the incoming DM particle hitting the DM target, in our case $s = m_s m_h$. With the integration bounds of Ref. [78] this leads to the typical scaling of the total rate with the DM mass

$$\sigma_{\langle s\rangle s \to \gamma\gamma} \approx \frac{\lambda_{hs}^2\, g_{h\gamma\gamma}^2}{4\pi} \frac{m_s}{m_h^3}\,. \tag{48}$$

We also need the local DM particle number density $n_{\text{DM}}$

$$n_{\text{DM}} = \frac{\rho_{\text{DM}}}{m_s} \approx \frac{10^{-41}}{m_s}\, \text{GeV}^4\,. \tag{49}$$

Inserting the maximum allowed value from Higgs to invisible searches for the Higgs portal coupling, $\lambda_{hs} = 8.7 \cdot 10^{-3}$, we arrive at a mean free path of

$$\lambda = \frac{4\pi}{\lambda_{hs}^2 g_{h\gamma\gamma}^2} \frac{m_h^3}{\rho_{\text{DM}}} \gtrsim 10^{43} \, \text{m} \,. \tag{50}$$

The crucial observation is that a light DM mass cancels between the cross section and the particle density. Hence, the mean free path for light DM particle ($m_s \ll m_h$) due to scattering at the DM background is universally bigger than the size of the observable universe, $l_{\text{univ}} \approx 30 \, \text{Gpc} \approx 10^{27} \, \text{m}$, by a factor of $10^{16}$. That's not good news for the LHC.

For slightly larger DM masses we can briefly look at the competing annihilation channel

$$\langle s \rangle s \to f\bar{f} \,, \tag{51}$$

with the cross section

$$\sigma_{\langle s \rangle s \to \bar{f}f} = \frac{\lambda_{hs}^2}{8\pi} \frac{m_f^2}{m_h^4} \left( 1 - \frac{4m_f^2}{m_s m_h} \right) \,, \tag{52}$$

where we again use $s = m_s m_h$ and assume $m_s \ll m_h$. This cross section is always positive above threshold, which for electrons is $m_s > 8.4 \, \text{eV}$, but it decreases for small DM mass. For electrons and $m_s \sim 10 \, \text{eV}$ the relic number density will become very small and the mean free path will still be $\lambda \approx 10^{39} \, \text{m}$, and the process hence unobservable.

To summarize, the mean free path of a dark matter particle in the DM background field with our local DM density can be written as

$$\lambda = 8.5 \, \text{m} \, \frac{m_s}{10^{-22} \text{eV}} \, \frac{10^{-6} \, \text{GeV}^{-2}}{\sigma_{\langle s \rangle s \to \text{sth.}}} \,. \tag{53}$$

The first term implies that the DM abundance increases with decreasing DM mass, and the mean free path decreases. The second term says that higher cross sections also shorten the mean free path. Applied to fuzzy DM with $m_s \sim 10^{-22} \, \text{eV}$ annihilating to $\gamma\gamma$, the first term becomes $\mathcal{O}(1)$ but due to the $m_s/m_h^3$ suppression in the cross section, we cannot even come close to the $10^{-6} \, \text{GeV}^{-2}$ of the numerator. For annihilations into $e^+e^-$ the cross section is larger but still suppressed by $m_e^2/m_h^4$, *i.e.* nowhere close to $10^{-6} \, \text{GeV}^{-2}$. In either case, we face a large Higgs mass suppression coming from the Higgs propagator in the annihilation process. The only way out of this is to introduce a new scale that compensates or replaces at least some suppression factors. Replacing the light scalar with a pseudoscalar does not help, either. Instead, it adds a momentum suppression relative to the Higgs mass which further reduces the rate.

## 3.2 Dark matter scattering

The second LHC process we explore is that of a light scalar produced in Higgs decays scattering with the detector material. A sufficiently light scalar will be highly energetic and can break up the nucleus in deep inelastic scattering. This will lead to a characteristic signature of a spontaneously appearing hadronic jet in the dense calorimeter material. This signature is inspired by direct detection of dark matter, even though in our case it would not confirm the dark matter nature of the light new scalar. The Feynman diagram for the process is shown in Fig. 6. The difference to the usual DIS process in high energy physics is that the nucleus $N$ is not highly relativistic. Instead, approximanting $E_s \approx m_h/2$, the center-of-mass energy of the scalar–nucleus scattering is $\sqrt{s} = \sqrt{2E_s M} \approx \sqrt{m_h M}$. As in the last section, we start with

the kinematics of our process before we compute the hard matrix element. To compute deep inelastic scattering of a scalar $s$ off the detector material of LHC experiments we will once more consider the mean free path,

$$\lambda = \frac{1}{n_{\text{det}} \, \sigma_{\text{DIS}}} \,, \tag{54}$$

of a light scalar $s$ in the detector material with the number density $n_{\text{det}}$. As an example we consider the ATLAS calorimeters. Both, the electromagnetic (ECAL) and hadronic calorimeter (HCAL) are sampling detectors using lead and iron as absorber materials [107]. As DIS takes place at the level of nucleons rather than the full nuclei we are interested in the nucleon density per unit target material. The effective nucleon density for a material $X$ can be computed as,

$$n_X = N_A \, \rho_X \, \frac{A_X}{m_X^{\text{mol}}} \,, \tag{55}$$

where $N_A$ denotes Avogadro's number, $A_X$ the mass number, $\rho_X$ the density, and $m_X^{\text{mol}}$ the molar mass of the material $X$. In the central region of the detector the ECAL has a radial extension of $L_E = 0.6$ m and the HCAL of $L_H = 2$ m. The inner tracking detector is a gas detector and hence can be neglected due to its low number density.

For each of the two detector materials we can compute the partonic cross section for the process shown in Fig. 6,

$$sN \rightarrow sg + X \,. \tag{56}$$

The partonic DIS cross section is usually given in terms of the energy loss $\nu$ of the scalar and the two kinematic variables $x = Q^2/(2M\nu)$ and $y = \nu/E_s$. The nucleon scattering cross section can then be expressed as the incoherent sum of partonic cross sections weighted by their respective parton distribution function,

$$\frac{d\sigma_{\text{DIS}}}{dx\,dy} = \sum_j \frac{d\hat{\sigma}_{\text{DIS}}}{dx\,dy} \, f_j(x, Q^2)\,. \tag{57}$$

As long as we only consider DM particles dominantly coupling to gluons, the incoherent sum in Eq.(57) reduces to weighting of the partonic cross section with the gluon PDF. To study DIS in the material of the LHC detectors we need to take into account nuclear effects by using the nuclear rather than proton PDF sets. Therefore, we perform all our calculations with the nCTEQ15 nuclear PDF set [108] via the ManeParse package [109]. As a cross check we compare the results with those using the MMHT proton densities [110]. For instance in case

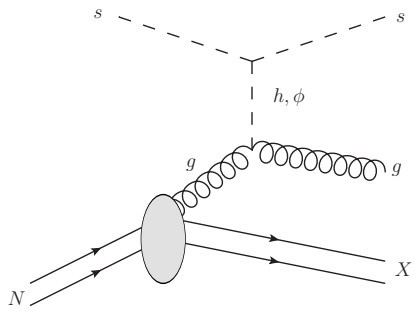

Figure 6: Deep inelastic scattering of a light scalar $s$ off a nucleon.

of a scalar mediator we find about 40% more events predicted by the appropriate proton PDF sets. Once we know the interaction rate $\sigma_{\text{DIS}}$ we can compute the probability that a single particle $s$ scatters in a detector of length $L_{\text{det}}$ as

$$P_{\text{DIS}} = 1 - e^{-\sum L/\lambda} \,. \tag{58}$$

**Scalar Higgs portal**   The differential cross section for the hard scattering process in the Higgs portal model is given by

$$\frac{d^2\hat{\sigma}_{\text{DIS}}}{dx\,dy} = \frac{\lambda_{hs}^2 g_{hgg}^2}{4\pi\hat{s}} \frac{Q^4}{(Q^2 + m_h^2)^2} \,, \tag{59}$$

where $\hat{s} = xs = 2ME_s x$ and $Q^2 = 2ME_s x y$. The loop-induced Higgs coupling to gluons, $g_{hgg} = \alpha_s/(12\pi)$, is defined in Eq.(4). This partonic cross section has to be convoluted with the gluon density in the heavy nucleus and integrated over the full phase space. In the case of the scalar Higgs portal the full DM DIS cross section on lead and iron evaluate numerically to

$$\sigma_{\text{Fe}} = 5.3 \cdot 10^{-9}\,\text{fb} \qquad \text{and} \qquad \sigma_{\text{Pb}} = 5.5 \cdot 10^{-9}\,\text{fb}\,. \tag{60}$$

The total scattering probability of a particle moving radially outwards is then given by

$$P_{\text{DIS}} = 1 - e^{-L_{\text{E}} n_{\text{Pb}} \sigma_{\text{Pb}}} e^{-L_{\text{H}} n_{\text{Fe}} \sigma_{\text{Fe}}} \approx 7.5 \cdot 10^{-21}\,. \tag{61}$$

We can combine this scattering probability with the Higgs production rate at the LHC and compute the expected number of dark matter DIS events for the maximum allowed branching ratio from supernova cooling constraints, $\lambda_{hs} \approx 2.5 \cdot 10^{-4}$, discussed in Sec. A.4,

$$N_{\text{DIS}} = \mathcal{L}_{\text{HL}}\, \sigma_h\, \text{BR}_{h \to ss}\, P_{\text{DIS}} \approx 4.1 \cdot 10^{-16}\,. \tag{62}$$

Inserting the Higgs production rate at $\sqrt{s} = 14\,\text{TeV}$ of around $\sigma_h \approx 60\,\text{pb}$ [111] and the total integrated luminosity expected in the high-luminosity run of the LHC (HL-LHC) of $\mathcal{L}_{\text{HL}} \approx 3\,\text{ab}^{-1}$, we find that this process is hopeless to observe in the renormalizable Higgs portal model.

**Scalar with new mediator**   A more flexible alternative to the renormalizable Higgs portal is a new scalar mediator $\phi$ with an effective coupling to gluons. Before we study the DIS signature at the LHC we note that such a mediator can decay into a pair of gluons or a pair of DM particles,

$$\Gamma_{\phi \to gg} = \frac{2\,\alpha_s^2}{\pi} \frac{m_\phi^3}{\Lambda_\phi^2}\,,$$

$$\Gamma_{\phi \to ss} = \frac{1}{32\pi} \frac{\mu_{\phi s}^2}{m_\phi} \sqrt{1 - 4\frac{m_s^2}{m_\phi^2}}\,. \tag{63}$$

This means that the coupling to gluons will always lead to a di-jet resonance $\phi \to gg$. At the LHC this resonance can be searched for in a bump hunt on top of a smoothly falling QCD di-jet background.

Even though the corresponding limits cannot be simply translated from narrow resonance searches to our model with a potentially wide mediator, we briefly report on the rough order of existing constraints. In the low-mass regime of $50 - 300$ GeV, CMS has looked for vector resonances $Z' \to \bar{q}q$ in di-jet events with an additional jet from initial state radiation (ISR) [112, 113]. In this search, the di-jet system from the $Z' \to \bar{q}q$ resonance has to recoil against a hard ISR jet, where at least one of the jets satisfies $p_T > 500$ GeV. This cut

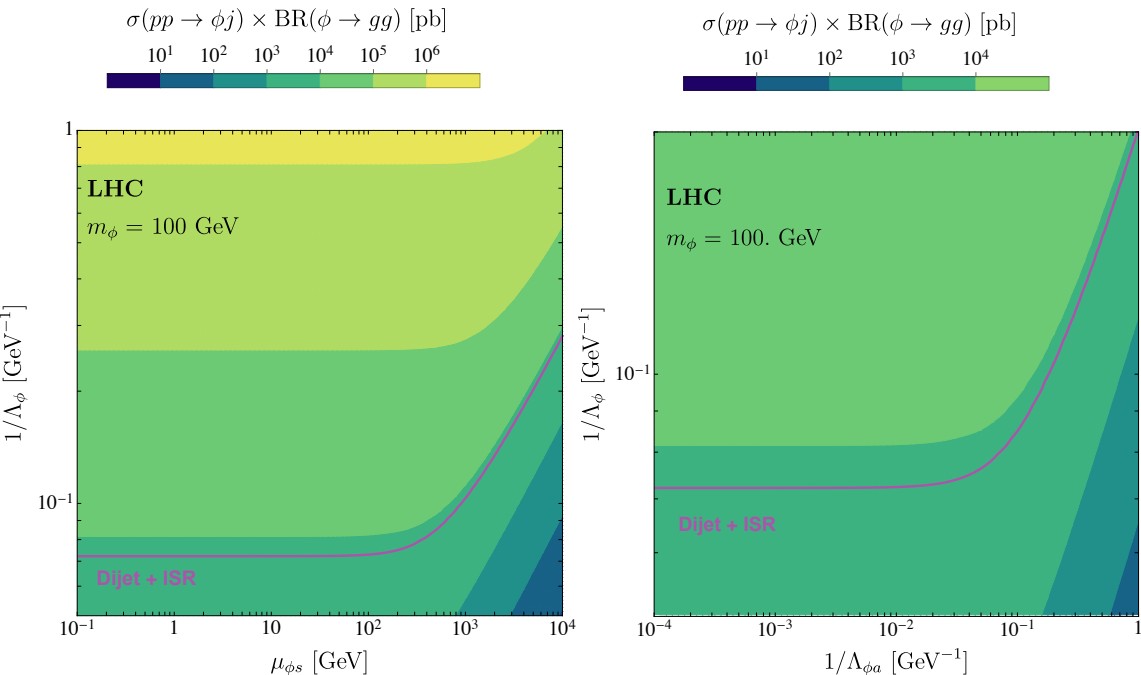

Figure 7: Cross section times branching ratio for the process $pp \to (\phi \to gg) + j$ for $m_\phi = 100$ GeV for the scalar (left) and pseudoscalar (right) model with a scalar mediator. The purple contour shows the CMS cross section limit of Ref. [112].

ensures that the low-mass $Z'$ is heavily boosted and the di-jet from the resonance will be reconstructed as a single large jet, back to back with the ISR jet. We can reinterpret this analysis for our mediator model in a simplistic way by generating the $(\phi \to gg) + j$ signal with just a cut of $p_T > 500$ GeV on the scalar. We generate the relevant signal events, $pp \to \phi + j$, with MadGraph5_aMC@NLO [114]. The corresponding cross section times branching ratio is shown in Fig. 7. The purple contour represents the CMS cross section limit for a 100 GeV mediator derived in Ref. [112]. Such a boosted jet analysis should be more stable for a broadening resonance than for instance a trigger-level resonance search. Nevertheless, we do not claim that an actual analysis for our model will ever be as good as the narrow-width $Z'$ search and only quote the CMS limits as the most optimistic estimate.

In further analogy to the Higgs portal case, the DM scalars $s$ can undergo DIS in the detector material via a mediator $\phi$. We can again calculate the partonic DIS cross section,

$$\frac{d^2\hat{\sigma}_{\mathrm{DIS}}}{dx\,dy} = \frac{\alpha_s^2}{4\pi\hat{s}} \left(\frac{\mu_{\phi s}}{\Lambda_\phi}\right)^2 \frac{Q^4}{(Q^2 + m_\phi^2)^2} \,. \tag{64}$$

To compute the total number of expected DIS events we simulate DM production, $pp \to \phi \to ss$, with MadGraph5_aMC@NLO [114] for a large range of mediator couplings to gluons ($1/\Lambda_\phi$) and couplings to the dark matter scalar ($\mu_{\phi s}$). The corresponding number of expected DIS events at the HL-LHC is shown in the left panel of Fig. 8 for a mediator of mass $m_\phi = 100$ GeV. The purple area is approximately excluded by the di-jet limit. In our Monte Carlo study we analyze the $p_T$- spectra of the produced DM particles $s$ and confirm that in the regime of a narrow mediator, $\Gamma_\phi/m_\phi \lesssim 10\%$, the averaged DM energy is $\langle E_s \rangle \approx 39$ GeV, with the bulk of the particles having $E_s = m_\phi/2 = 50$ GeV. To compute the number of DIS events we assume this average energy over all displayed parameter space. This is conservative, as in the case of a broader resonance the produced DM particles become more energetic on average, thus enhancing the DIS cross section. In a fully-fletched analysis the scattering cross section should be

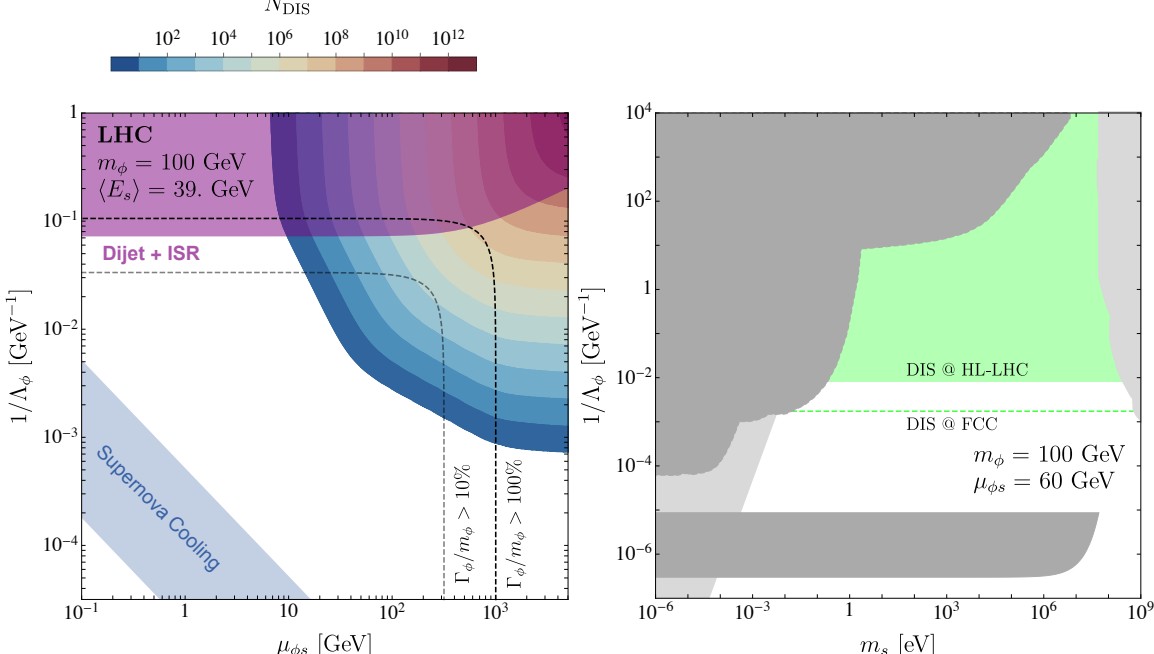

Figure 8: Left: number of expected DIS events in the plane of DM-mediator coupling $\mu_{\phi S}$ versus mediator-gluon coupling $\Lambda_\phi$ for $m_\phi = 100$ GeV. The blue band represents the constraint from Supernova cooling, the purple area is the bound on low-mass di-jet resonances [112], and the dashed lines indicate fixed ratios $\mu_{\phi S}/\Lambda_\phi$. Right: comparison of the projected DIS reach with the low-energy constraints in terms of $\Lambda_\phi$ for fixed $\mu_{\phi s}$. Constraints which require the dark matter nature are shown in light grey.

convoluted with the DM energy spectrum. However, this is beyond the scope of this sensitivity study. Similarly, we assume that the displaced recoil jet signature is essentially background-free. At the LHC this statement is never strictly true, because for instance detector failures or support structures can of course generate displaced objects. Moreover, if a highly energetic jet were to consist only of long-lived neutral hadrons it could generate such a recoil, but such a strong suppression of all charged hadrons is rather unlikely.

As always, passing the LHC triggers is the first challenge for our signal. Barring other trigger opportunities we can rely on the standard mono-jets trigger requiring missing transverse momentum around 100 GeV. We emphasize that in addition to the standard mono-jets signature we can use the displaced recoil jet to reduce the SM-backgrounds. On the other hand, there might be additional handles on the trigger, so we will quote projected limits without the trigger requirements. We have checked that a for a momentum-independent scalar coupling the trigger catches at least 10% of the signal rate and weakens the projected limits in the coupling $1/\Lambda_\phi$ by at most a factor three.

In the right panel of Fig. 8 we contrast the projected reach of the DIS process with the low-energy and other limits from Fig. 3. We see that the DIS probe is complementary to all other constraints and fills the gap for dark matter masses between 1 eV and 100 MeV over two orders of magnitude in the gluon coupling $1/\Lambda_\phi$. The projected FCC sensitivity corresponds to a collider energy of 100 TeV and a luminosity of $\mathcal{L} = 30$ ab$^{-1}$ [115]. More details on the FCC estimate can be found in App. A.6. In terms of parameter reach the FCC projections exceed the HL-LHC projections by another order of magnitude in the coupling.

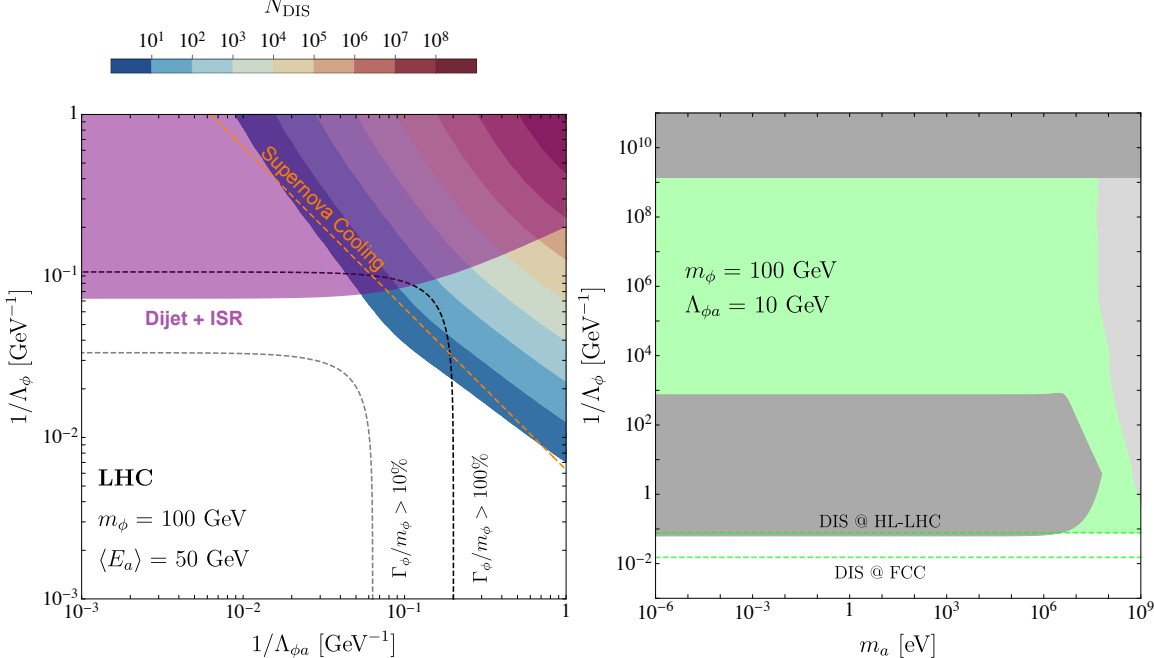

Figure 9: Left: number of expected DIS events in the plane of DM-mediator coupling $\Lambda_{\phi a}$ versus mediator-gluon coupling $\Lambda_{\phi}$ for $m_{\phi} = 100$ GeV. Right: comparison of the projected DIS reach with the low energy constraints in terms of $\Lambda_{\phi}$ for fixed $\Lambda_{\phi a} = 10$ GeV. Constraints which require the dark matter nature are shown in light grey.

**Pseudoscalar with Higgs mediator**   The third model we consider for the DIS signature is ALP dark matter with a derivative Higgs portal. Repeating the previous calculation for the analogous process with the scalar $s$ replaced by the shift-symmetric scalar $a$ yields the differential DIS cross section

$$\frac{d^2\hat{\sigma}_{\text{DIS}}}{dx\,dy} = \frac{g_{hgg}^2}{16\pi\hat{s}}\,\frac{Q^4}{\Lambda_{ha}^4}\left(\frac{Q^2 + 2m_a^2}{Q^2 + m_h^2}\right)^2 , \tag{65}$$

where $\hat{s} = xs = 2ME_a x$ and $Q^2 = 2ME_a x y$. We always assume the minimum suppression scale from the Higgs to invisible limit of $\Lambda_{ah} \approx 832$ GeV. Integrating this cross section in the limit $m_a \ll m_h$ and using once more the ATLAS calorimeter materials and dimensions as a benchmark, the full DM DIS cross section on lead and iron give us

$$\sigma_{\text{Pb}} = 5.7 \cdot 10^{-12}\,\text{fb} \qquad \text{and} \qquad \sigma_{\text{Fe}} = 6.5 \cdot 10^{-12}\,\text{fb} . \tag{66}$$

This means that a single produced pseudoscalar $a$ undergoes DIS in the detector with a probability of

$$P_{\text{DIS}} \approx 8.8 \cdot 10^{-24} . \tag{67}$$

The total number of expected DIS events is given by

$$N_{\text{DIS}} \approx 2.2 \cdot 10^{-16} . \tag{68}$$

As for the ULDM scalar with the Higgs portal, this process is unobservable at the HL-LHC.

**Pseudoscalar with new mediator**     In the same spirit as for the scalar case, we again consider the simplified model for the pseudoscalar $a$ with a new scalar mediator. As the shift-symmetric pseudoscalar couples only via derivatives, its interaction strengths are in general momentum-dependent. At the LHC $a$ is produced from the decaying mediator $\phi$, which again is produced in gluon fusion. As in the scalar case, we generate the corresponding signal, $pp \to \phi \to aa$, with MadGraph5_aMC@NLO [114]. Our Monte Carlo study shows that as long as the mediator has a comparatively narrow width, $\Gamma_\phi / m_\phi \lesssim 10\%$, the produced DM particle $a$ carries roughly half the mediator mass in momentum. For a weak-scale mediator mass $m_\phi \sim v$ this is typically enough such that $a$ can undergo deep inelastic scattering with the nuclei in the detector material. The relevant cross section of the hard scattering process,

$$aN \to ag + X \, , \tag{69}$$

reads

$$\frac{d^2 \hat{\sigma}_{\mathrm{DIS}}}{dx \, dy} = \frac{\alpha_s^2}{16\pi \hat{s}} \frac{Q^4}{\Lambda_{\phi a}^2 \Lambda_\phi^2} \left( \frac{Q^2 + 2m_a^2}{Q^2 + m_\phi^2} \right)^2 . \tag{70}$$

The left panel of Fig. 9 displays the expected number of DIS events at an ATLAS-like detector for the high luminosity run of LHC. The grey and black dashed lines show the contours of $\Gamma_\phi / m_\phi \lesssim 10\%$ and $\lesssim 100\%$. The effect of the mono-jets trigger is significantly smaller than for the scalar case, because of the momentum-dependent DM-coupling. We estimate the trigger survival rate of the pseudoscalar signal to be above 70%, translating into a negligible 15% shift in the coupling reach.

Again, the right panel of Fig. 9 shows the corresponding HL-LHC and FCC projections in model space, compared to all other limits from Fig. 4. The DIS signature closes the wide gap from all other limits over 13 orders of magnitude in $m_a$ and covers the supernova constraints, providing an independent collider probe of the cosmological observations. The FCC projections again exceed the HL-LHC projections by an order of magnitude in the coupling and provides the leading signatures for pseudoscalar dark matter with a weak-scale mediator.

# 4   Conclusions

Light dark matter is a relatively new avenue of dark matter model building and phenomenology. It leads us to consider a wealth of new measurements and cosmological observations, while challenging established search strategies like large-scale direct detection of hadron collider analyses. It also forces us to go back and forth between a semi-classical wave-like description and a quantum field theory Lagrangian.

We have studied models with a light scalar or pseudoscalar dark matter agent with a mass ranging from well below the eV scale to the GeV scale with a focus on bridging the quasiclassical and quantum descriptions. As mediators to the Standard Model we have assumed the SM-like Higgs or a new, weak-scale scalar. We have studied a large number of constraints from low-energy precision measurements, big bang nucleosynthesis, supernova cooling, invisible Higgs decays, and direct dark matter detection. The relative impact of these constraints depends strongly on the quantum numbers of the dark matter and on the nature of the mediator. While BBN strongly constrains very light scalar dark matter and invisible Higgs decays obviously only apply to models with a SM-like Higgs mediator, supernova constraints are very model independent. Current direct detection experiments, like Xenon1T, start cutting into the parameter space at relatively large dark matter masses. The difference between scalar and pseudoscalar dark matter can largely be understood by the derivative interactions of the pseudoscalar and its effect on the interaction rates.

Inspired by the process topologies of direct and indirect detection we have studied two novel LHC signatures. On the one hand, light dark matter particles produced for instance in Higgs decay can annihilate with the DM background in the LHC detector. In analogy to indirect detection, but with the benefit of probing the actual dark matter property, we can look for pairs of photons or electrons produced in the LHC detectors. Unfortunately, we find the rate of this signal to be very low. On the other hand, similar to direct detection in nucleon recoils, light dark matter produced at the LHC can hit the nuclei in the ATLAS and CMS calorimeters and produce a *hard, displaced recoil jet*. This completely new signature of ULDM should be observable for scalar mediator models and over the full range of dark matter masses below the GeV scale. For scalar ULDM this signature closes a gap in all current constraints from $m_s = 1$ eV to the direct detection thresholds around $m_s = 100$ MeV. For pseudoscalar ULDM it closes the entire gap between atomic spectroscopy measurements and supernova cooling for $m_a = 10^{-6} \dots 10^7$ eV, again all the way to large-scale direct detection experiments. Unlike direct detection signals this LHC search does not assume a relic density, and the field-theoretical description allows for a consistent comparison to cosmological supernova observations. In that sense it can also be immediately generalized to other very light particles, for instance neutrinos, as long as they can be produced at the LHC with sufficient energy and rate.

## Acknowledgments

We thank Joerg Jaeckel, Felix Kling, Simon Knapen, Ennio Salvioni and Hans-Christian Schultz-Coulon for useful discussions. MB and PF are funded by the UK Science and Technology Facilities Council (STFC) under grant ST/P001246/1. TP is supported by the DFG Transregio *Particle physics phenomenology after the Higgs discovery* (TRR 257). PR is funded by the DFG Graduiertenkolleg *Particle physics beyond the Standard Model* (GRK 1940).

## A Calculational details

In this Appendix we collect details of the various computations performed to derive the various limits in this paper.

### A.1 Variation of fundamental constants

Measurements that are sensitive to variations in the fundamental constants $m_f, \alpha$ and $m_V$ set stringent constraints on models including scalars and pseudoscalars for ULDM with masses below $m_s \ll$ eV. SM-like operators that describe the fermion masses $m_f$, the coupling $\alpha$, and the gauge boson masses $m_V$ are

$$\mathcal{L}_{\text{SM}} \supset -\sum_f m_f \bar{f} f - \frac{F_{\mu\nu} F^{\mu\nu}}{4} + \sum_V \delta_V m_V^2 V_\mu V^\mu \,, \tag{71}$$

with $\delta_W = 1$ and $\delta_Z = 1/2$. For instance in the Higgs portal model of Eq.(3) these constants are modified by the effective interaction operators

$$\mathcal{L} \supset \frac{\lambda_{hs}}{2} \frac{m_f}{m_h^2} s^2 \bar{f} f - \frac{\lambda_{hs} g_{h\gamma\gamma}}{2} \frac{1}{m_h^2} s^2 F_{\mu\nu} F^{\mu\nu} - \lambda_{hs} \delta_V \frac{m_V^2}{m_h^2} s^2 V_\mu V^\mu \,. \tag{72}$$

For the scalar model of Eq.(33) with a new mediator the corresponding effective Lagrangian reads

$$\mathcal{L} \supset -\frac{m_a^2 m_f}{2\Lambda_{ha}^2 m_h^2} a^2 \bar{f} f + \frac{g_{h\gamma\gamma}}{2} \frac{m_a^2}{\Lambda_{ha}^2 m_h^2} a^2 F_{\mu\nu} F^{\mu\nu} + \delta_V \frac{m_a^2 m_V^2}{\Lambda_{ha}^2 m_h^2} a^2 V_\mu V^\mu \,. \tag{73}$$

Wherever we treat the DM field as a classical field, we can simply translate these two sets of Lagrangians into each other via $\lambda_{hs} s^2 \leftrightarrow -m_a^2/\Lambda^2 a^2$. In both the scalar and pseudoscalar case, we insert the classical field solution and rewrite the quadratic (pseudo-) scalar interaction in a constant and a time-dependent part as done in Ref. [15]

$$s^2 = s_0^2 \cos^2(m_s t) \to \frac{s_0^2}{2} (1 + \cos(2m_s t))$$

$$a^2 = a_0^2 \cos^2(m_a t) \to \frac{a_0^2}{2} (1 + \cos(2m_a t)) \,. \tag{74}$$

In this form the constant term describes a fifth force while the oscillating terms lead to a variation of fundamental constants, for instance the fermion mass

$$m_f \to m_f \left[ 1 + \frac{s^2}{\Lambda_{s,f}^2} \right] = m_f \left[ 1 + \frac{s_0^2}{2\Lambda_{s,f}^2} + \frac{s_0^2}{2\Lambda_{s,f}^2} \cos(2m_s t) \right], \tag{75}$$

with $1/\Lambda_{s,f}^2 = \lambda_{hs}/(2m_h^2)$. The variation of the fine structure constant and the weak boson masses can be derived in complete analogy with $1/\Lambda_{\gamma,s}^2 = 2\lambda_{hs} g_{h\gamma\gamma}/m_h^2$ and $1/\Lambda_{s,V}^2 = \lambda_{hs}/m_h^2$ and accordingly for the derivative case.

## A.2 Multi-pseudoscalar exchange

We mention in the main text that a pseudoscalar field $a$ obeying a shift symmetry can be coupled to SM fields via derivative couplings. For long-range forces mediated by the exchange of $a$ there are important differences between linear couplings of $a$ to the SM and theories in which only operators with multiple $a$ insertions feature. For a linear derivative coupling of $a$ to an axial current,

$$\frac{\partial_\mu a}{2f} \bar{\psi}\gamma^\mu\gamma^5\psi + \frac{\partial_\mu a}{2f} \bar{\chi}\gamma^\mu\gamma^5\chi \,, \tag{76}$$

long-range forces between the fermions $\psi$ and $\chi$ can be induced due to $s$-channel pseudoscalar exchange. The corresponding amplitude reads

$$\overline{|\mathcal{M}|^2} = \frac{1}{64f^4} \frac{1}{(q^2 - m_a^2)^2} \text{Tr}[(\not{p}_2 - m_\psi)\not{q}\gamma^5(\not{p}_1 + m_\psi)\not{q}\gamma^5] \text{Tr}[(\not{p}_3 + m_\chi)\not{q}\gamma^5(\not{p}_4 - m_\chi)\not{q}\gamma^5]$$

$$= \frac{m_\psi^2 m_\chi^2}{f^4 \left(1 - 2\frac{m_a^2}{q^2} + \frac{m_a^4}{q^4}\right)} = \frac{m_\psi^2 m_\chi^2}{f^4} + \mathcal{O}\left(\frac{m_a^2}{q^2}\right). \tag{77}$$

In the limit of large momentum transfer $q \gg m_a$ the leading term of the amplitude is $q$-independent.

The leading operator for derivative couplings with two $a$ insertions is given by (33),

$$\frac{\partial_\mu a \, \partial^\mu a}{2\Lambda_{ha}^2} H^\dagger H \,.$$

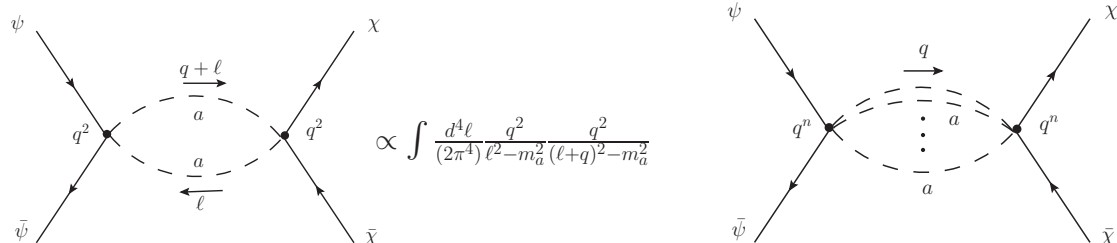

Figure 10: Left: Two-to-two scattering via two-pseudoscalar exchange. Right: Two-to-two scattering via $n$-pseudoscalar exchange.

At low energy we can integrate out the Higgs such that the derivative Higgs portal induces interactions of the type

$$\mathcal{L} \supset \frac{1}{v} \frac{\partial_\mu a \partial^\mu a}{2\Lambda_{ha}^2} \bar{\psi}\psi + \frac{1}{v} \frac{\partial_\mu a \partial^\mu a}{2\Lambda_{ha}^2} \bar{\chi}\chi . \tag{78}$$

Similar to the case of linear derivative interactions this leads to the scattering shown in Fig. 10. The corresponding matrix element reads

$$i\mathcal{M} = \bar{v}(p_2) \frac{iq^2}{2v\Lambda_{ha}^2} u(p_1) \int \frac{d^4\ell}{(2\pi^4)} \frac{i}{\ell^2 - m_a^2} \frac{i}{(\ell + q)^2 - m_a^2} \ \bar{u}(p_3) \frac{iq^2}{2v\Lambda_{ha}^2} v(p_4)$$

$$\propto q^4 \int dx \log\left(\frac{x\Lambda_c}{m_a^2 - x(1-x)q^2}\right), \tag{79}$$

where $\Lambda_c$ is a momentum cutoff of the loop integral. Obviously, the resulting amplitude $\overline{|\mathcal{M}|^2}$ has no $q$-independent part and vanishes $\propto q^4$ at low momentum transfer. Operators with additional derivatives increase the power of the momentum $q$ associated with the $n$-point vertex. In contrast, the overall momentum transfer flowing through the diagram will always be equal to $q^2$. Any long-range force mediated in such a theory is momentum-suppressed.

Theories with multi-pseudoscalar exchange are therefore qualitatively different from theories with a linearly coupled pseudoscalar. The sensitivity of experiments with small momentum exchange is strongly suppressed in the case of multi-pseudoscalar exchange which makes the case for complementary approaches beyond astrophysical and precision measurements of low-energy observables.

### A.3 Big bang nucleosynthesis

In the derivation of the BBN limits in Section 2.1 and Section 2.2 we express the helium yield as in (15) and [15]

$$\frac{\Delta(n/p)_W}{(n/p)_W} = -0.13\frac{\Delta\alpha}{\alpha} - 2.7\frac{\Delta(m_d - m_u)}{(m_d - m_u)} - 5.7\frac{\Delta M_W}{M_W} + 8.0\frac{\Delta M_Z}{M_Z} ,$$

$$\frac{\Delta\Gamma_n}{\Gamma_n} = -1.9\frac{\Delta\alpha}{\alpha} + 10\frac{\Delta(m_d - m_u)}{(m_d - m_u)} - 1.5\frac{\Delta m_e}{m_e} + 10\frac{\Delta M_W}{M_W} - 14\frac{\Delta M_Z}{M_Z} . \tag{80}$$

In the case of a simplified model where only couplings to gluons are present, Eq. (15) becomes particularly simple and one can write [116]

$$\frac{\Delta Y}{Y} = \left(-\frac{Q_{np}}{T_W} + t_{\text{BBN}} \frac{\partial_x P(x)}{P(x)}\right) \frac{\Delta Q_{np}}{Q_{np}} \approx 4.82 \frac{\Delta Q_{np}}{Q_{np}} , \tag{81}$$

where $P(x)$ is the phase space in the neutron decay width [116]. The energy density of a non-relativistic oscillating DM field is given by $\rho \simeq m_s^2 \langle s^2 \rangle$ and evolves according to

$$\bar{\rho}_{DM} = 1.3 \cdot 10^{-6}[1+z(t)]^3 \frac{\text{GeV}}{\text{cm}^3}, \tag{82}$$

with the redshift parameter $z(t)$. For a non-oscillating DM field, we have $\rho \simeq m_s^2 \langle s^2 \rangle / 2$ and

$$\bar{\rho}_{DM} = 1.3 \cdot 10^{-6}[1+z(t_m)]^3 \frac{\text{GeV}}{\text{cm}^3}, \tag{83}$$

with $z(t_m)$ defined by $H(t_m) \approx m_s$. In both cases, we assume that the mean DM energy density during weak freeze-out is much greater than the present-day local cold DM energy density $\langle \phi^2 \rangle_{\text{weak}} \gg \langle \phi^2 \rangle_0$. In the case of the oscillating field, we make use of the relation $[1+z(t_m)]/(1+z_{\text{weak}}) = \sqrt{t_{\text{weak}}/t_m}$ and take

$$
\begin{aligned}
t_{\text{weak}} &\approx 1.1 \text{ s} \\
z_{\text{weak}} &\approx 3.2 \cdot 10^9 \\
H(t_m) &\sim 1/(2t_m) \sim m_s \rightarrow t_m \sim 1/(2m_s)
\end{aligned}
\tag{84}
$$

from [15].

## A.4 Supernova energy loss

Stars can be used as a particle-physics laboratory by studying the energy-loss rate implied by new low-mass particles such as ULDM particles. Any annihilation process from SM to light new particles contributes to supernova cooling. The main assumption in the corresponding calculation is that the produced particles can freely escape the supernova, the so-called free-streaming limit. It allows us to set an upper bound on the coupling strength of the additional processes. New particles only cause significant effects if they can compete with the cooling from neutrinos already carrying away energy directly from the interior of stars. The strongest bound comes from the SN1987A [81],

$$\varepsilon_x < 10^{19} \text{ erg g}^{-1}\text{s}^{-1}. \tag{85}$$

To set constraints, one has to evaluate the novel energy-loss rates at typical core conditions with a temperature and density of around

$$T_{\text{SN}} = 30 \text{ MeV} \qquad \text{and} \qquad \rho_{\text{core}} = 3 \cdot 10^{14} \text{ g cm}^{-3}, \tag{86}$$

or directly set limits on the total energy-loss rate per unit mass,

$$\Gamma = \varepsilon_x \rho_{\text{core}} < 10^{-14} \text{ MeV}^5. \tag{87}$$

After being produced, new particles travel through the SN core and might start interacting with the supernova. The process considered here is elastic scattering $Ns \rightarrow Ns$ with a $t$-channel Higgs or scalar particle exchange. To estimate for which couplings light scalars start to interact with the supernova particles, we compare the radius of the supernova, $R_{\text{SN}} \approx 10$ km, with the mean free path of elastic scattering,

$$\lambda = \frac{1}{n_N(r)\,\sigma_{sN \rightarrow sN}}, \quad n_N(r) = \begin{cases} \dfrac{\rho_{\text{core}}}{m_p} & \text{for } r \leq R_{\text{SN}}, \\[2ex] \dfrac{\rho_{\text{core}}}{m_p}\left(\dfrac{R_{\text{SN}}}{r}\right)^m & \text{for } r > R_{\text{SN}}, \end{cases} \tag{88}$$

and $m = 3 \dots 7$ depending on the profile chosen. This condition characterizes the trapping regime, *i.e.* the point where the new scalars start thermalizing and are trapped sucht that they cannot escape the supernova freely anymore. Once the scalars are trapped they create a scalarsphere similar to the axiosphere [81,84]. In the regime where the free-streaming limit doesn't apply anymore, the sphere still looses energy via black-body-radiation. The radius of the sphere $r_0$ can be determined by

$$4\pi r_0^2 \, \frac{g \pi^2}{120} \, T(r_0)^4 < 10^{53} \text{ ergs}^{-1} \, , \tag{89}$$

with a temperature profile of $T(r) = T_{\text{SN}}(R/r)^{m/3}$ [83] and $g = 1$ the number of effective degrees of freedom. It varies between

$$r_0 = 1.7 \dots 7.2 \cdot 10^6 \text{ cm} \, , \tag{90}$$

depending on $m$. The second condition for the black-body radiation of a scalarsphere with radius $r_0$ is an upper bound on the optical depth

$$\int_{r_0}^{\infty} \lambda^{-1} \mathrm{d}r \le \frac{2}{3} \, . \tag{91}$$

Combining the optical depth criterion with the upper bound on the luminosity of the scalarsphere, we will set a bound on the couplings in our models.

**Scalar dark matter**  Following Ref. [82], we consider two processes for the energy-loss rate in the free-streaming limit, photon annihilation $\gamma\gamma \to ss$ and the bremsstrahlung-like process $NN \to NNss$ both depicted in Fig. 11. Pair annihilations of photons yields a cross section of

$$\sigma_{\gamma\gamma \to ss} \approx \frac{\lambda_{hs}^2 g_{h\gamma\gamma}^2}{\pi} \, \frac{\omega^2}{m_h^4} \, , \tag{92}$$

where $\omega$ is the energy of the incoming photon. For the total energy loss for a thermalized gas of photons, we obtain [82]

$$\Gamma_{\gamma\gamma} = n_\gamma^2 \langle 2\omega\sigma_{\gamma\gamma \to ss} \rangle = \frac{32\pi\zeta(3)}{63} \lambda_{hs}^2 g_{h\gamma\gamma}^2 \, \frac{T_{\text{SN}}^9}{m_h^4} \, , \tag{93}$$

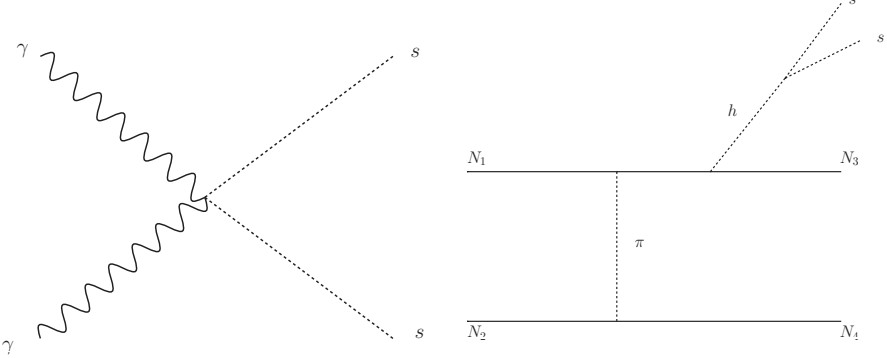

Figure 11: Photon annihilation (left) and bremsstrahlung-like scalar emission in nucleon-nucleon interactions. For the simplified model, the Higgs has to be replaced by the new mediator $\phi$ in the right plot.

where

$$n_\gamma = \frac{2\zeta(3)}{\pi^2} T_{\text{SN}}^3 \, ,$$

$$\langle 2\omega\, \sigma_{\gamma\gamma\to ss} \rangle = \frac{\int_0^\infty d\omega\, \omega^2\, 2\omega \sigma_{\gamma\gamma\to ss}/[\exp(\omega/T_{\text{SN}})-1]}{\int_0^\infty d\omega\, \omega^2\, 1/[\exp(\omega/T_{\text{SN}})-1]} \, . \tag{94}$$

With the temperature given in Eq.(86) we can translate Eq.(87) into

$$\lambda_{hs} < 0.17 \, . \tag{95}$$

The second process $NN \to NNss$ is calculated with an effective nucleon-scalar coupling. We can simply translate limits on $M_*$ in Ref. [82],

$$\langle E\sigma v \rangle \sim \frac{1}{12\pi^4} \frac{T^3 m_N^2}{M_*^4} \left(\frac{T}{m_N}\right)^{1/2} \sigma_{NN}$$

$$\Rightarrow \qquad \Gamma_{NN\to NNss} \sim \sigma_{NN\to NNss} \frac{n_N^2 T^{7/2} m_N^{3/2}}{12\pi^4 M_*^4} < 10^{-14}\,\text{MeV}^5 \, , \tag{96}$$

through $c_{sNN} = m_N/(2M_*^2)$ into the model parameters for scalar ULDM

$$\lambda_{hs} < 2.75 \cdot 10^{-4} \qquad\qquad \text{(Higgs portal)} \, ,$$

$$\frac{\mu_{\phi s}}{\Lambda_\phi} < 1.8 \cdot 10^{-5} \left(\frac{M_S}{100\,\text{GeV}}\right)^2 \quad \text{(scalar mediator)} \, . \tag{97}$$

For the Higgs portal we see that the constraints from $NN \to NNss$ scattering are stronger than those from photon annihilation. In the following, we therefore omit the annihilation process.

For the trapping limit we assume the nuclei in the supernova to be at rest, because $T_{SN} \sim 30$ MeV $\ll M_N$. The elastic scattering cross section is given by

$$\sigma_{sN\to sN} = \frac{1}{8\pi}\, c_{sNN}^2\, \frac{2 + 4\frac{E_s}{M_N} + \frac{E_s^2}{M_N^2} + \frac{m_s^2}{M_N^2}}{\left(1 + 2\frac{E_s}{M_N} + \frac{m_s^2}{M_N^2}\right)^2} \simeq \frac{1}{4\pi}\, c_{sNN}^2 \, , \tag{98}$$

where we assume $m_\phi \ll M_N$ as well as $E_\phi \sim T_{SN} \ll M_N$. The coupling strength $c_{sNN}$ for the scalar models can be found in Eq.(10). With that cross section, we find a constraint on the coupling and suppression scale through the optical depth criterion

$$\lambda_{hs} > 6.7 \dots 8.1 \cdot 10^{-3} \qquad\qquad \text{(Higgs portal)} \, ,$$

$$\frac{\mu_{\phi s}}{\Lambda_\phi} > 4.3 \dots 5.2 \cdot 10^{-4} \left(\frac{m_\phi}{100\,\text{GeV}}\right)^2 \quad \text{(scalar mediator)} \, . \tag{99}$$

For all limits in the trapping regime, we state a range for the constraint that is coming from the variable choice of $m$ in the temperature and density profile. In our final plots, we choose the trapping limit that maximizes the excluded bands for supernova constraints.

**Pseudoscalar dark matter** For the derivative portal models, we start with the energy-loss rate from Ref. [117]

$$\Gamma_{NN\to NNaa} = \varepsilon_x \rho_{\text{core}} \simeq \frac{264\sqrt{\pi}}{\pi^4}\left(3 - \frac{2\beta}{3}\right) n_B^2 \left(\frac{g_N m_N}{m_r^2 m_h^2}\right)^2 \alpha_\pi^2 \frac{T^{9.5}}{m_N^{4.5}} \, , \tag{100}$$

where $\alpha_\pi = (2m_N f/m_\pi)^2/4\pi$ with $f \approx 1$ is the pion-nucleon "fine-structure" constant, $g_N m_N/(m_r^2 m_h^2)$ is the effective Higgs-nucleon coupling and $\beta$ is a nucleon-momentum dependent term with $\beta = 2.0938$ [117]. We replace the effective $\alpha N$-coupling with the effective $aN$-coupling $g_N m_N/(m_r^2 m_h^2) \to c_{aNN}$, and obtain the limits on our model parameters

$$\frac{1}{\Lambda_{ha}} < \frac{0.3}{\text{GeV}} \qquad\qquad\qquad\qquad \text{(Higgs mediator)},$$

$$\frac{1}{\Lambda_\phi} < \frac{6.3 \cdot 10^{-2}}{\text{GeV}} \left(\frac{\Lambda_{\phi a}}{10\,\text{GeV}}\right)\left(\frac{m_\phi}{100\,\text{GeV}}\right) \quad \text{(scalar mediator)}. \tag{101}$$

For the squared matrix element in the derivative simplified model, we get

$$|\mathcal{M}|^2 = \frac{1}{8} c_{aNN}^2 (2m_a^2 - t)^2 (4m_N^2 - t). \tag{102}$$

By assuming $m_a \ll E_a \ll m_N$, this yields a cross section of

$$\sigma_{aN \to aN} = \frac{c_{aNN}^2}{12\pi} T^4, \tag{103}$$

where $c_{aNN}$ is taken from Eq. 37. Following the calculations of the scalar case, the limits in the derivative portal models are

$$\frac{1}{\Lambda_{ha}} > 17.1 - 34.1\,\text{GeV}^{-1},$$

$$\frac{1}{\Lambda_\phi} > 2.0 - 7.4 \cdot 10^2\,\text{GeV}^{-1} \left(\frac{\Lambda_{\phi a}}{10\,\text{GeV}}\right)\left(\frac{m_\phi}{100\,\text{GeV}}\right)^2. \tag{104}$$

## A.5 Direct detection

In all our DM scenarios the scalar DM candidate couples to nucleons via a scalar mediator, either the SM Higgs $h$ or a new singlet $\phi$. At low energies these interactions will lead to elastic DM-nucleus scattering, where the incoming DM particle will transfer part of its momentum to the nucleus and generate a nuclear recoil. The corresponding spectrum depends on the DM-nucleon scattering cross section [118–120]. In all cases where the DM-nucleon interactions carry a scalar Lorentz structure, the induced scattering is spin-independent.

**Scalar Higgs portal**   In a first step we derive the relevant effective low-energy operator. For the scalar Higgs portal it is generated when we integrate out the Higgs in Eq.(3) and take into account the Higgs-quark coupling $m_q/v\, h\, \bar{q}q$,

$$\mathcal{L} \supset \frac{\lambda_{hs}\, m_q}{2\, m_h^2} s^2\, \bar{q}q. \tag{105}$$

To obtain the scattering cross section of the DM particle with nucleons we evaluate the matrix element of the effective (partonic) operator with asymptotic nucleon states,

$$\mathcal{M} = \langle n(k')|\mathcal{O}_{\text{eff}}|n(k)\rangle. \tag{106}$$

For light quarks we evaluate the nuclear matrix element [121],

$$\langle n(k')|m_q\, \bar{q}q|n(k)\rangle = m_n\, f_{T,q}^n\, \bar{u}_n(k') u_n(k), \tag{107}$$

while for heavy quarks we will integrate out the heavy quark fields via the QCD trace anomaly [120], leading for each heavy quark field to the replacement

$$m_q \bar{q}q \rightarrow -\frac{\alpha_s}{12\pi} \operatorname{Tr} G_{\mu\nu} G^{\mu\nu}. \tag{108}$$

The nuclear matrix element can then be evaluated via

$$\langle n(k')|\alpha_s \operatorname{Tr} G_{\mu\nu} G^{\mu\nu}|n(k)\rangle = -\frac{8\pi}{9} m_n f_{T,g}^n \bar{u}_n(k') u_n(k). \tag{109}$$

The coefficients $f_{T,q}^n$ and $f_{T,g}^n$ parametrize the nuclear matrix elements and can be found *e.g.* in Table II of Ref. [120]. Summing over all the quarks, the full matrix element for the DM-nucleon scattering reads

$$\mathcal{M} = \frac{\lambda_{hs}}{m_h^2} m_n \left( f_{T,u}^n + f_{T,d}^n + f_{T,s}^n + \frac{2}{9} f_{T,g}^n \right) \bar{u}_n(k') u_n(k). \tag{110}$$

To turn this into a cross section section for DM-nucleon scattering we perform a non-relativistic expansion of the matrix element. This is a good approximation as long as the scattered nuclei are at rest and the DM particles in the local halo are non-relativistic. In the non-relativistic limit the we can take the normalization of the Dirac spinors as $u(k)^s = \sqrt{m}\,(\xi_s, \xi_s)^T$, where $\xi_s$ are the two-component Weyl spinors normalized such that $\xi_s^\dagger \xi_s = 1$. This allows us to write the non-relativistic nuclear matrix element as

$$\mathcal{M}_{NR} = \frac{\lambda_{hs} m_n}{m_h^2} \left( f_{T,u}^n + f_{T,d}^n + f_{T,s}^n + \frac{2}{9} f_{T,g}^n \right) (2m_n) \xi_{s'}^\dagger \xi_s. \tag{111}$$

Summing and averaging over the nucleon spins we arrive at the squared matrix element,

$$|\mathcal{M}|^2 = \left( \frac{2\lambda_{hs} m_n^2}{m_h^2} \right)^2 \left( f_{T,u}^n + f_{T,d}^n + f_{T,s}^n + \frac{2}{9} f_{T,g}^n \right)^2. \tag{112}$$

If the squared matrix element is independent of the scattering angle we can trivially integrate over the scattering angle to obtain the total scattering cross section

$$\sigma_{sn \rightarrow sn} = \frac{1}{\pi} \left( \frac{\mu_{sn} \lambda_{hs} m_n}{2 m_s m_h^2} \right)^2 \left( f_{T,u}^n + f_{T,d}^n + f_{T,s}^n + \frac{2}{9} f_{T,g}^n \right)^2, \tag{113}$$

with the reduced DM-nucleon mass $\mu_{sn} = m_s m_n/(m_s + m_n)$.

**Scalar with new mediator**   The case of a scalar DM particle $s$ coupling to a new scalar mediator $\phi$ described by Eq (5) differs mainly by the interactions of $\phi$ with matter. In contrast to the Higgs portal model, the interaction is mediated via the effective coupling to gluons rather than the renormalizable couplings to quarks. We find the relevant effective operator at energy scales much below the mediator mass $m_\phi$ to be

$$\mathcal{L} \supset \frac{\alpha_s \mu_{\phi s}}{2\Lambda_\phi m_\phi^2} s^2 \operatorname{Tr} G_{\mu\nu} G^{\mu\nu}. \tag{114}$$

Using the result for the nuclear matrix element of the gluon operator from Eq.(109) we can write down the nuclear-level matrix element for the DM-gluon contact term,

$$\mathcal{M} = -\frac{8\pi}{9} \frac{m_n \mu_{\phi s}}{\Lambda_\phi m_\phi^2} f_{T,g}^n \bar{u}_n(k') u_n(k). \tag{115}$$

Repeating the same steps as for the Higgs portal model the DM-nucleon elastic scattering cross section becomes

$$\sigma_{sn \rightarrow sn} = \frac{16\pi}{81} \left( \frac{\mu_{sn} m_n \mu_{\phi s}}{\Lambda_\phi m_s m_\phi^2} \right)^2 (f_{T,g}^n)^2. \tag{116}$$

**Pseudoscalar with Higgs mediator**   When the pseudoscalar $a$ couples to the Higgs via the derivative portal of Eq.(33), integrating out the Higgs yields a momentum-dependent contact term,

$$\mathcal{L} \supset \frac{m_q}{m_h^2 \Lambda_{ha}^2} \frac{p \cdot p'}{2} a^2 \bar{q} q . \tag{117}$$

The corresponding nuclear matrix element for DM-nucleon scattering with asymptotic nucleon initial and final states is

$$\mathcal{M} = \frac{p \cdot p'}{m_h^2 \Lambda_{ha}^2} m_n \left( f_{T,u}^n + f_{T,d}^n + f_{T,s}^n + \frac{2}{9} f_{T,g}^n \right) \bar{u}_n(k') u_n(k) . \tag{118}$$

In terms of the reduced mass $\mu_{an} = m_a m_n / (m_a + m_n)$ we arrive at the full expression for total cross section in the non-relativistic limit,

$$\sigma_{na \to na} = \frac{\mu_{an}^2 m_a^2 m_n^2}{4\pi \Lambda_{ah}^4 m_h^4} \left[ 1 - 2 \left( 1 - \frac{\mu_{an}}{m_a} \right) v_a^2 + \left( 1 - 2 \frac{\mu_{an}}{m_a} + \frac{4}{3} \frac{\mu_{an}^2}{m_a^2} \right) v_a^4 \right]$$

$$\times \left( f_{T,u}^n + f_{T,d}^n + f_{T,s}^n + \frac{2}{9} f_{T,g}^n \right)^2 , \tag{119}$$

where the higher power terms in $v_a \sim 10^{-3}$ can be ignored.

**Pseudoscalar with new mediator**   Lastly, in the case of a derivative portal coupling to a new scalar mediator, Eq.(35), we obtain the low-energy contact term

$$\mathcal{L} \supset -\frac{\alpha_s}{\Lambda_{\phi a} \Lambda_\phi m_\phi^2} \frac{p \cdot p'}{2} a^2 \operatorname{Tr} G_{\mu\nu} G^{\mu\nu} . \tag{120}$$

Evaluating the matrix element with asymptotic nucleon initial and final states as before and performing a non-relativistic expansion we obtain the scattering rate

$$\sigma_{na \to na} = \frac{16\pi}{81} \frac{\mu_{an}^2 m_a^2 m_n^2}{\Lambda_{ah}^2 \Lambda_\phi^2 m_h^4} \left[ 1 - 2 \left( 1 - \frac{\mu_{an}}{m_a} \right) v_a^2 + \left( 1 - 2 \frac{\mu_{an}}{m_a} + \frac{4}{3} \frac{\mu_{an}^2}{m_a^2} \right) v_a^4 \right] \left( f_{T,g}^n \right)^2 . \tag{121}$$

## A.6   FCC projections

Complementing our LHC analysis of DIS in scalar simplified DM models at the LHC in Sec. 3.2, we illustrate our results for a future circular hadron collider (FCChh) with an energy of 100 TeV and an integrated luminosity of 30 ab$^{-1}$ [115]. As for the LHC we generate Monte Carlo events using `MadGraph5_aMC@NLO` [114] for the DM production channels

$$pp \to \phi \to ss , aa . \tag{122}$$

The mediator $\phi$ couples to gluons through the usual dimension-5 operator. For these signals we calculate the number of expected DIS events as outlined in Sec. 3.2. We illustrate our result in Fig. 12. Comparing our findings for the FCC in the simplified portal model in the left panel of Fig. 12 with those for the LHC shown in the left panel of Fig. 8, we see that at the FCC we expect roughly a factor of hundred more events for a given point in the parameter space. For the pseudoscalar, derivative case we compare the right panel of Fig. 12 to the left panel of Fig. 9 and find a thousand times as many events for the FCC. This is expected from the momentum enhancement shown in Eq.(64) compared to Eq.(70).

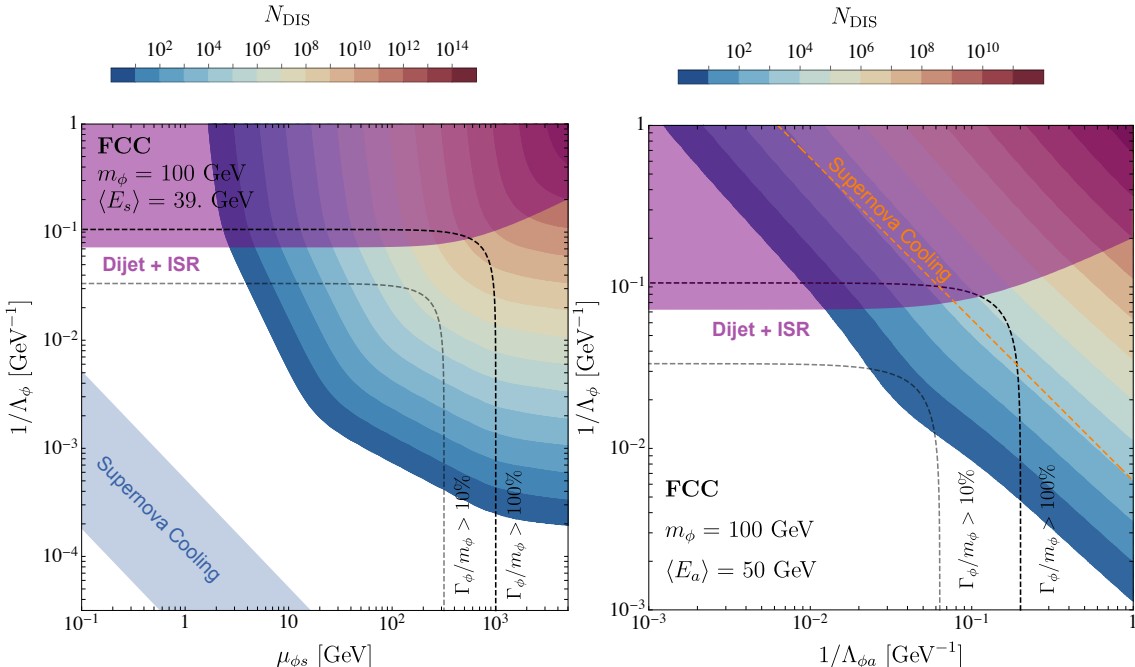

Figure 12: Left: number of expected DIS events at the FCChh in the plane of DM-mediator coupling $\mu_{\phi s}$ versus mediator-gluon coupling $\Lambda_\phi$ for scalar ULDM with $m_\phi = 100$ GeV. Right: same for the pseudoscalar ULDM.

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
