# Peer review of "Light Dark Matter Annihilation and Scattering in LHC Detectors"

_SciPost Physics, doi:SciPost Phys. 10, 030 (2021)_

## Round 2 · Referee Report · Suchita Kulkarni · 2020-8-23

Strengths
The paper explores novel dark matter signatures at the LHC. The strategy is to follow direct detection philosophy at the LHC, where the DM produced via collisions can recoil off detector material or background DM density.
It provides detailed calculations of several constraints for scalar and Goldstone boson dark matter interacting with the SM via Higgs and scalar portal.
The paper in general targets light dark matter at the LHC using novel signatures. In particular, it considers DM production via Higgs or scalar mediator and such DM then either scatters off with background DM or initiates a DIS process with nucleus in detector material. It shows that a large range of DM masses can be targeted using the technique described in the paper. This is new and interesting way of exploring light DM at the LHC.
Report
First of all, I apologies for the delay in the report. I hope the authors are not too uncomfortable with the delay.
I recommend the publication of this paper in the journal. In general the presentation is good, the calculations are clear and several formulae are presented in details. This is something many times absent in other papers and hence very much appreciated. I have a few minor questions as given below. I request the clarification of those.
Requested changes
1) It is possible that for sub-GeV DM the cosmic ray propagation can be affected, see https://arxiv.org/abs/1810.07705. Do the authors know whether such effects are relevant for their analysis as well?
2) The authors do not consider DM - electron scattering signatures when analysing non-LHC constraints. Is this because they are suppressed by scalar couplings?
3) For the DM interactions with background density as described in figure 5, I suppose additional complications will arise since parts of the detector are solid. The best location to target such background density would be inside the beam pipe. Which in turn would mean that only certain scalar mean free paths can be targeted. This of course does not change the analysis which already shows that scattering with background density will in general will not be favourable, however this is a comment out of curiosity.
4) Have the authors considered the effect of similar analysis at low energy
and fixed target experiments? The interactions will of course loose in cross section but gain in luminosity. In addition, a different detector material plus different mediator masses can be targeted.
(in reply to Report 2 on 2020-09-10)
In the attached file we have addressed the issues raised by the referees and listed the according changes we have made to the paper in the resubmitted version.
Attachment:
report.pdf

---

## Round 2 · Referee Report · Anonymous · 2020-9-10

Report
Referee report to the paper on Light Dark Matter Annihilation and Scattering at the LHC Detectors
By Martin Bauer et al.
The paper covers a discussion of ultra-light dark matter (DM) down to the lowest possible masses, and proposes a novel detection method for detecting scalar and pseudoscalar dark matter, would it be produced at the LHC.
The idea of observing produced DM in the LHC experiments though annihilation with the omnipresent DM halo, or via inelastic scattering of the DM with the detector material, is quite intriguing. While no usable event rate was found for the first process, the second can allow for covering an interesting region of phase space for Dark Matter searches which the experimentalists need to be made aware of, since the required analyses are somewhat atypical (but doable) in the experiment. In addition there is a very nice first part of the paper where limits for ultra light DM are discussed in great detail, giving useful summary plots of the status of the field for this search.
The analysis methods used in this paper, input data and assumptions, are sound and appear to be complete. The paper is extremely well written and therefore I recommend it for acceptance in the journal.
I have a few questions to the work, which I believe are non-critical but probably can be answered quickly by the authors.
- Figure 2: the left and right plot
These plots have the same information (to my understanding) with a corresponding color code for the excluded regions, but are shown for two different ranges of the m_s and lambda parameter I was a bit surprised to see such a large exclusion region on the right plot from the Eot-Wash experiment limits, when on the left side plot the importance for these limits seems to stop already on 10^-8 in mass. Can that be explained?
- The DM produced in the Higgs/scalar decays is high energetic due to the mass of resonance mass around 100 GeV. If these particles make up the DM hallo, do we have an estimate of the energy spectrum these particles can have? If they can be high energetic other large volume precision detectors (even near surface) could perhaps look for these? I am thinking on e.g. the new generation of kiloton or multi-kiloton liquid argon detectors that start to get operational. Of course such detectors need to deal with neutrino backgrounds as the signal is not time related with a high energy collision as for the LHC.
- plots 8-9 are for a 100 GeV mediator. The mass of such a putative mediator is not known. Can one in indicate how the phase space coverage changes within some limits (eg factor 10 up and down).
- microcosmic
last paragraph of the conclusion "the the "

---

## Round 3 · Author Response

We have addressed the issues raised by the referees in a separate PDF file, which is attached as a reply to the referee report.

---

## Round 3 · List of Changes

As also outlined in our reply PDF we have additionally added a paragraph concerning potential contributions to N_eff in the early universe of light (pseudo-)scalar DM.

---

## Editorial Decision

published